# 📖 MMIE: Massive Multimodal Interleaved Comprehension Benchmark for Large Vision-Language Models

**Peng Xia[1]***,**Siwei Han[1]***, **Shi Qiu[1]***, **Yiyang Zhou[1], Zhaoyang Wang[1], Wenhao Zheng[1],**
**Zhaorun Chen[3], Chenhang Cui[4], Mingyu Ding[1], Linjie Li[2], Lijuan Wang[2], Huaxiu Yao[1]**
[1]UNC-Chapel Hill, [2]Microsoft Research, [3]University of Chicago, [4]NUS
{pxia,siweih,shiqiu,huaxiu}@cs.unc.edu

## Abstract

Interleaved multimodal comprehension and generation, enabling models to produce and interpret both images and text in arbitrary sequences, have become a pivotal area in multimodal learning. Despite significant advancements, the evaluation of this capability remains insufficient. Existing benchmarks suffer from limitations in data scale, scope, and evaluation depth, while current evaluation metrics are often costly or biased, lacking in reliability for practical applications. To address these challenges, we introduce MMIE, a large-scale knowledge-intensive benchmark for evaluating interleaved multimodal comprehension and generation in Large Vision-Language Models (LVLMs). MMIE comprises 20K meticulously curated multimodal queries, spanning 3 categories, 12 fields, and 102 subfields, including mathematics, coding, physics, literature, health, and arts. It supports both interleaved inputs and outputs, offering a mix of multiple-choice and open-ended question formats to evaluate diverse competencies. Moreover, we propose a reliable automated evaluation metric, leveraging a scoring model fine-tuned with human-annotated data and systematic evaluation criteria, aimed at reducing bias and improving evaluation accuracy. Extensive experiments demonstrate the effectiveness of our benchmark and metrics in providing a comprehensive evaluation of interleaved LVLMs. Specifically, we evaluate eight LVLMs, revealing that even the best models show significant room for improvement, with most achieving only moderate results. We believe MMIE will drive further advancements in the development of interleaved LVLMs. We publicly release our benchmark and code in https://mmie-bench.github.io/.

Content warning: this paper contains content that may be inappropriate or offensive.

## 1 Introduction

*"True evaluation lies in the seamless interweaving of diverse modalities."*

Multimodal learning has made remarkable progress with the development of Large Vision-Language Models (LVLMs) (Liu et al., 2023a; Zhu et al., 2023; Dai et al., 2023), which are capable of handling diverse tasks that involve both images and text. Despite their advancements, most of these models are limited to multimodal tasks for text generation, such as visual question answering (VQA) and image captioning, which do not fully reflect the potential of multimodal capacity. To broaden their application, interleaved text-and-image generation has emerged as a critical area of research (Liu et al., 2024). It requires models to generate images and text in any sequence, thereby enhancing the versatility and effectiveness of multimodal systems. It opens up possibilities for various complex applications, such as multi-step inference (Lu et al., 2024; Kazemi et al., 2024), multimodal situational analysis (Yang et al., 2021), and visual storytelling (Huang et al., 2016).

While recent LVLMs are evolving to support interleaved text-and-image generation (Team, 2024; Xie et al., 2024; Chern et al., 2024; Zhou et al., 2024), a comprehensive evaluation benchmark is still falling behind due to the following two challenges:

---

*Equal Contribution.

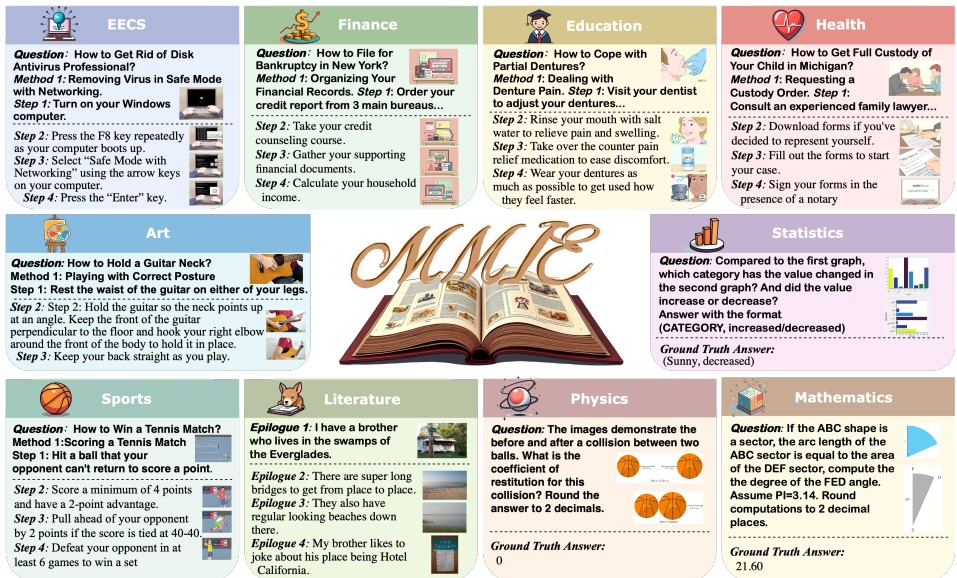

Figure 1: Typical samples from the MMIE Benchmark showcase its support for multiple image inputs and outputs, with ground truth provided for every query. MMIE evaluates models across diverse fields, ensuring a comprehensive evaluation of their capabilities.

- **Difficulty in Constructing Modality-Coherent Benchmarks.** The first challenge lies in the difficulty of constructing modality-aligned multimodal datasets, where both the input and output contain images and text. Current benchmarks mainly focus on single-modality output tasks (Fu et al., 2023; Li et al., 2024a; Zhang et al., 2023), assessing only the quality of the generated image or text, without benchmarking the crucial connection between modalities, such as text-image coherence and consistency. Although a few datasets support the interleaved multimodal evaluation method for LVLMs (Liu et al., 2024), their dataset is constrained by its limited scale and narrow query format, primarily focused on VQA tasks.

- **Lack of Automated Evaluation Metric.** The second challenge is the lack of suitable automated evaluation metrics for interleaved generation. Human evaluation is costly and time-consuming, making it difficult to scale for practical applications. Current automated evaluation metrics typically assess either the quality of generated text (e.g., BLEU (Papineni et al., 2002), BERTScore (Zhang et al., 2020)) or the quality of generated images (e.g., FID (Heusel et al., 2017)). While recent evaluation strategies, such as using CLIPScore (Hessel et al., 2021), and vision-language models (VLMs) (Chen et al., 2023; Liu et al., 2024), can evaluate the connection between different modalities, they rely heavily on the pre-trained knowledge of specific models (e.g., CLIP training data) or follow rigid, human-defined rules. These approaches can introduce bias and uncertainty to some extent, often leading to inconsistent results (Mahmoud et al., 2024).

To address these limitations, we introduce **MMIE**, a **M**assive **M**ultimodal **I**nverleaved understanding **E**valuation benchmark for LVLMs with proposed reliable and automated metrics. MMIE is curated from four multimodal datasets, involving 3 categories, 12 fields, and 102 subfields, including mathematics, physics, coding, statistics, literature, philosophy, education, finance, health, sports, art, and EECS (Electrical Engineering and Computer Science). The dataset comprises 20K multimodal questions, supporting both interleaved inputs and outputs. It features a mix of multiple-choice and open-ended question formats to evaluate a broad spectrum of competencies across various fields. As shown in Table 2, MMIE surpasses existing interleaved multimodal benchmark in both depth and width, particularly in addressing complex problem-solving and open-ended creative tasks. Based on the curated dataset, we further propose an automated metric powered by a scoring model. Specifically, we first design a comprehensive evaluation criteria for each category. Then, we curate a fine-grained, human-annotated scoring dataset and then use this dataset to fine-tune the InternVL-2 (Chen et al., 2024c) to obtain the scoring model. Using MMIE, we evaluate four open-source interleaved multimodal LVLMs, as well as combinations of advanced LVLMs like GPT-4o with

text-to-image generative models (e.g., Stable Diffusion 3 (Esser et al., 2024)). Our key contributions are summarized as follows:

- We introduce the largest high-quality interleaved multimodal benchmark MMIE for evaluating LVLMs, with the dataset to be publicly released.

- MMIE presents significant challenges to LVLMs, with the best-performing model (e.g., GPT-4o + SDXL) achieving a score of 65.47%, highlighting substantial room for improvement.

- The proposed scoring model is reliable and has proven to be comparable to human evaluation.

## 2 RELATED WORK

**Interleaved Multimodal Comprehension and Generation.** Multimodal learning has rapidly evolved, with substantial progress in integrating text and image modalities. Recent advancements in large vision-language models (LVLMs) (Liu et al., 2023a; Zhu et al., 2023; 2024; Dai et al., 2023; Xia et al., 2024b;c), either driven by the integration of diffusion models like Stable Diffusion (Rombach et al., 2022), or using token-based mixed-modal structures like Chameleon (Team, 2024) and Show-o (Xie et al., 2024), have enabled models to not only understand and generate content across modalities, but also engage in interleaved multimodal comprehension and generation. As the demand for richer, more interactive AI grows, interleaved multimodal comprehension and generation is becoming an essential component in the development of next-generation LVLMs.

**LVLM Benchmarks.** Despite the rapid advancements in multimodal learning, evaluation benchmarks remain far from perfect. Previous benchmarks primarily focused on evaluating the base perception ability of LVLMs (Lu et al., 2022; Gurari et al., 2018), such as GQA (Hudson & Manning, 2019), which lack the depth required to assess advanced reasoning. Recently, several high-quality evaluation benchmarks have been proposed to assess the reasoning ability of these models (Li et al., 2024a; Zhang et al., 2023; Liu et al., 2023a;b; Yu et al., 2023; Xia et al., 2024a; Jiang et al., 2024b; Zhang et al., 2024b;b;c; Jiang et al., 2025), such as MMMU (Yue et al., 2024) and MME (Fu et al., 2023). However, these benchmark do not support interleaved image-and-text comprehension and generation. Large-scale interleaved multimodal datasets like MINT-1T (Awadalla et al., 2024), MANTIS (Jiang et al., 2024a) and OBELICS (Laurençon et al., 2024) have been developed primarily for pre-training models. However, they lack precise alignment between text and images, making them unsuitable for evaluation and benchmarking. A recent small-scale interleaved multimodal benchmark has been introduced (Liu et al., 2024), but its limited data size and query quality hinder the comprehensiveness of its evaluation. MMIE fills this gap by offering a comprehensive evaluation framework that supports interleaved multimodal comprehension and generation. Our dataset includes a diverse set of queries among multiple domains. By evaluating both perceptual and generative capacity of LVLMs, it provides a more holistic assessment.

**Evaluation Metrics for Multimodal Tasks.** Traditional evaluation metrics, such as BLEU (Papineni et al., 2002), BERTScore (Zhang et al., 2020) for text quality, and FID (Heusel et al., 2017) for image quality, are only suited to single-modality output tasks. Recent metrics, such as CLIPScore (Hessel et al., 2021) and X-IQE (Chen et al., 2023), have attempted to address this by introducing multimodal models to evaluate consistency between text and image. However, these metrics only measure alignment and fall short of offering a comprehensive assessment of output quality. Furthermore, many multimodal metrics depend on GPT-based models (Liu et al., 2024), bringing uncontrollable bias to the whole evaluation

Table 1: Dataset statistics.

| Statistic | Number | Percentage |
|---|---|---|
| Total questions | 20103 | - |
| - Situational analysis | 5005 | 24.89% |
| - Project-based learning | 11482 | 57.12% |
| - Multi-step reasoning | 3616 | 17.99% |
| Total Categories/Fields/Subfields | 3/12/102 | - |
| Formats: | | |
| - Multiple-Choice Questions | 663 | 3.40% |
| - Open-Ended Questions | 19340 | 96.60% |
| Questions with Images | 20103 | 100% |
| Questions with answer label | 20103 | 100% |
| Average question length | 76.0 | - |
| Average images per question | 1.32 | - |

system. To overcome these drawbacks, we propose an automatic metric to minimises bias and provides a thorough analysis of the generated results.

# 3 THE MMIE BENCHMARK

## 3.1 OVERVIEW

In this section, we introduce MMIE, a diverse and comprehensive benchmark for evaluating interleaved multimodal comprehension and generation across a broad scope of tasks. As shown in Table 2, MMIE consists of 20,103 curated samples spanning 12 fields, including mathematics, physics, coding, statistics, literature, philosophy, education, finance, health, sports, art, and EECS. Each query is meticulously selected, filtered, and refined to ensure both high quality and relevance across the covered subjects. In addition, MMIE emphasizes the evaluation of three essential competencies: perception, reasoning, and generation. Unlike previous benchmarks that evaluate the results from single modality (Fu et al., 2023; Yue et al., 2024; Li et al., 2024b) output, MMIE is specifically designed to assess models' capabilities in understanding and generating

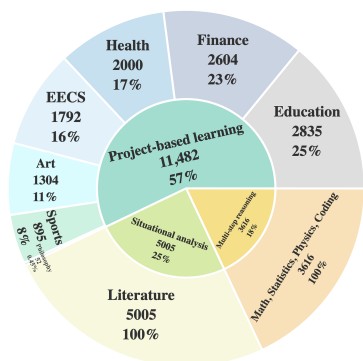

Figure 2: Distribution of categories and fields in MMIE.

interleaved text and images in any sequence. This evaluation extends beyond basic perception by requiring models to engage in complex reasoning, leveraging subject-specific knowledge across different modalities.

## 3.2 DATASET CURATION

The data curation process in MMIE consists of two stages, each designed to ensure both comprehensive coverage and high-quality representation across various categories in our benchmark. We detail the process as follows:

In the first stage, we collect and restructure four multimodal datasets to align with the interleaved image-and-text format and categorize them into three categories – situational analysis, project-based learning and multi-step reasoning, which are illustrated in Figure 2. Specifically, for project-based learning, we extract data from Wikihow (Yang et al., 2021), which is originally designed for testing models' ability to choose the correct procedural steps based on given text and image contexts. We adapt it to the interleaved text-and-image format. For situational analysis, we draw samples from VIST (Huang et al., 2016), a naturally interleaved multimodal dataset designed for visual storytelling tasks, which challenges models to seamlessly integrate narrative text and images. Both situational analysis and project-based learning datasets feature in-

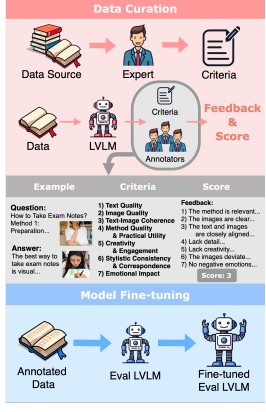

Figure 3: Pipeline of the scoring model.

terleaved inputs and outputs. To expand the benchmark with more complex and diverse tasks, we further introduce datasets focused on multi-step reasoning that support interleaved inputs. For this aspect, we source examples from MathVista (Lu et al., 2024) and ReMI (Kazemi et al., 2024), which together provide 3,600 questions covering topics from functions to statistics. The answer formats for these queries include multiple-choice questions (selecting one option from several choices) and open-ended questions (directly generating content). After extracting samples from these four datasets, we merge and refine them into a cohesive benchmark by compacting, restructuring, and integrating questions from multiple sources, ensuring consistency with our evaluation objectives.

In the second stage, we implement a multi-step quality control process to ensure the integrity and consistency of the dataset. First, we apply lexical overlap and source URL similarity checks to identify and flag potential duplicate entries, which are then manually reviewed and removed. Next, each dataset is meticulously reviewed for formatting and typographical consistency to ensure adherence to a standardized structure. Discrepancies are corrected to maintain uniformity across the entire dataset. In total, we finally collect 20,103 instances across 12 fields, including mathematics, physics, coding, statistics, literature, philosophy, education, finance, health, sports, art, and EECS. Detailed categorization and dataset statistics are presented in Table 1. For more information about dataset curation, please refer to Appendix A.1.

Table 2: The comparison between MMIE with other LVLM benchmarks. Inter-I: interleaved input; Inter-O: interleaved output; Multi-I: multi-image for input; Multi-O: multi-image for output.

| Dataset | Data Scale | Inter-I | Inter-O | Multi-I | Multi-O | #Num Domains | Answer Type | Metric |
|---|---|---|---|---|---|---|---|---|
| HumanEval (Chen et al., 2021) | 164 | No | No | No | No | 1 | Open | Pass@k |
| GSM8K (Cobbe et al., 2021) | 8.5K | No | No | No | No | 1 | Open | Pass@k |
| MME (Fu et al., 2023) | 2K | Yes | No | No | No | 4 | Multi-Choice | ACC |
| MMBench (Liu et al., 2023b) | 3K | Yes | No | No | No | 6 | Multi-Choice | ACC |
| MM-Vet (Yu et al., 2023) | 218 | Yes | No | No | No | 6 | Open | GPT-4 |
| MagicBrush (Zhang et al., 2023) | 10K | Yes | No | No | No | 7 | Image Editing | CLIPScore |
| MMMU (Yue et al., 2024) | 11.5K | Yes | No | Yes | No | 30 | Multi-Choice | ACC |
| MVBench (Li et al., 2024b) | 4K | Yes | No | Yes | No | 9 | Multi-Choice | ACC |
| INTERLEAVEDBENCH (Liu et al., 2024) | 815 | Yes | Yes | Yes | Yes | 10 | Open | GPT-4o |
| **MMIE (Ours)** | **20K** | Yes | Yes | Yes | Yes | 12 | Multi-Choice & Open | Fine-tuned VLM |

## 3.3 AUTOMATED EVALUATION METRIC

As traditional metrics such as BLEU, BERTScore, and CLIP-Score fail to provide a thorough evaluation of the quality of multimodal outputs, existing benchmarks use the GPT-4 series as the scoring model, which may introduce inherent bias in the scoring process (Liu et al., 2024). To ensure a comprehensive and unbiased evaluation of various LVLMs, as shown in Figure 3, we propose an automated evaluation metric powered by our fine-tuned LVLM to assist in scoring. Here, we choose InternVL-2-4B (Chen et al., 2024c) as the foundation for our scoring system due to its strong performance in multimodal reasoning tasks and support for multi-image inputs. Furthermore, we fine-tune the InternVL-2-4B to mitigate potential bias.

Specifically, we first construct a high-quality multimodal scoring dataset that covers all aspects of our benchmark, accompanied by detailed scoring criteria and reference answers. In this process, we collect 800 responses from four LVLMs—MiniGPT-5 (Zheng et al., 2023), EMU-2 (Sun et al., 2024), GILL (Koh et al., 2023), and Anole (Chern et al., 2024). Based on the ground-truth labels, we define an evaluation standard using a six-point grading scale with clear criteria. A group of experts generates reference answers for each level and all score statistics are converted to percentage format. These criteria and reference answers together form a robust rubric for MMIE. Following the rubric, human annotators rigorously score the responses. Detailed examples of the rubric and construction process are provided in Appendix A.9 and Appendix A.3.

After constructing the scoring dataset, we fine-tune the InternVL-2-4B model and use the fine-tuned version as our scoring model. To validate its performance, we randomly select 200 new samples with human-scored labels and compare the results of our model with those of other scoring models. The results show that the fine-tuned model significantly improves alignment between human scores and our model-generated scores compared to other LVLMs, leading to more accurate and reliable evaluation across diverse tasks. We will discuss the experimental results in detail in Section 4.3.

## 3.4 COMPARISON WITH EXISTING MULTIMODAL BENCHMARKS

MMIE surpasses existing benchmarks in three key aspects. First, most previous multimodal benchmarks support only single-modality input or output, while MMIE closes this gap by enabling interleaved text-and-image comprehension and generation. Our dataset ensures robust modality alignment, with multimodal question-answer pairs reconstructed into an interleaved text-and-image instruction format, followed by manual review to guarantee quality. Moreover, the scenarios reflect real-world applications, such as multimodal script generation, data chart analysis, and multimodal story generation. Second, compared to recent interleaved comprehension benchmarks (Liu et al., 2024), MMIE is larger in scale and covers a broader range of subjects, containing both reasoning and temporal understanding skills, allowing for a more comprehensive evaluation. Finally, MMIE introduces a reliable scoring system powered by a fine-tuned LVLM, which significantly enhances the accuracy and reliability of scoring. Table 2 highlights the differences between our benchmark and existing ones, demonstrating the advantages of MMIE in terms of scale, diversity, and scoring methodology.

## 4 EXPERIMENT

MMIE provides a systematic evaluation of existing open-source LVLMs supporting interleaved multimodal input and output (**interleaved LVLMs**), along with the integration of state-of-the-art

LVLMs and text-to-image generative models (**integrated LVLMs**). In this section, we aim to answer the following key questions: (1) Which interleaved LVLM performs best on MMIE overall? (2) How effective are the integrated LVLMs? (3) Do the evaluated LVLMs show a preference for a certain field? and (4) How useful are our proposed model-powered metric compared with traditional metrics and other LVLM evaluation?

## 4.1 EXPERIMENT SETUP

**Baseline Models.** We first benchmark four open-source interleaved LVLMs. (1) MiniGPT-5 (Zheng et al., 2023), a multimodal model combining MiniGPT-4 and Stable Diffusion, specialized for coherent image-text generation. (2) EMU-2 (Sun et al., 2024), a 37B-parameter model excelling in in-context learning and multimodal reasoning, (3) GILL (Koh et al., 2023), a model specialized in generating and retrieving interleaved outputs, (4) Anole (Chern et al., 2024), based on Chameleon (Team, 2024), a model excelling in text quality, adds vision and multimodal generation capabilities.

To broaden the comparison, we also compare with integrated LVLMs consisting of text-output LVLMs (i.e., GPT-4o (Achiam et al., 2023), Gemini-1.5 (Reid et al., 2024), LLaVA-v1.6-34b (Liu et al., 2023a) and Qwen-VL-2-72b (Wang et al., 2024)) and text-to-image generative models (i.e., Openjourney (ope), Stable Diffusion 3 Medium (Esser et al., 2024), Stable Diffusion XL turbo, Flux.1-dev (flu)). We provide the interleaved text-and-image input to the LVLM to generate text, and then feed this text to a text-to-image generative model to generate an image. The resulting multimodal output from this process is considered as interleaved output for evaluation.

**Human Annotators.** We organize a group of senior top-tier college students, contributing to the curation of the scoring dataset. To ensure thorough and consistent evaluations, we develop detailed criteria for each category of our benchmark (see Appendix A.9 for details).

**Evaluation Metrics.** We evaluate the performance of all models using our proposed metric in Section 3.3, which is powered by our fine-tuned LVLM based on InternVL-2-4B (Chen et al., 2024c), to ensure reliable scoring.

## 4.2 MAIN RESULTS

In this section, we present the comprehensive evaluation on our MMIE benchmark. The detailed performance of interleaved LVLMs and integrated LVLMs is shown in Table 3 and Table 4, respectively. We summarize our key findings as follows:

**Challenging Evaluation and Promising Direction.** As illustrated in Table 3, all evaluated interleaved LVLMs show poor performance, with an average score of 50.80%. Even when integrating advanced models such as GPT-4o and text-to-image generative models, as shown in Table 4, the best score (GPT-4o + SDXL) reached is 65.47%. This highlights the high level of difficulty and the challenge posed by MMIE. Interestingly, the latest interleaved LVLM Anole (Chern et al., 2024) shows significant improvements over previous interleaved LVLMs, including MiniGPT-5, GILL and EMU-2, by 8.4%, 7.0%, 21.8% in average score, respectively. This points to the growing potential of interleaved text-and-image models as a promising direction for future progress in multimodal comprehension and generation. To facilitate the broader adoption of MMIE, we extract 1,000 samples to create a mini-set. Detailed results can be found in Appendix A.5.

**Gap between Interleaved LVLMs and Integrated LVLMs.** Existing interleaved LVLMs are still quite limited. To enhance our evaluation and analysis on our benchmark, we integrate non-interleaved LVLMs with T2I models in our experiments. This integrated LVLMs approach significantly outperforms previous open-source interleaved LVLMs, improving performance by an average of 25.2% across all categories. Specifically, the integrated models outperform the best performance of the interleaved model by 14.6%, 26.3%, and 16.1% in situational analysis, project-based learning, and multi-step reasoning, respectively. Surprisingly, the integrated LVLMs perform exceptionally well in project-based learning, with all models based on LLaVA-34b achieving scores above 70%. These findings suggest that combining the strong comprehension abilities of non-interleaved LVLMs with the generative power of T2I models offers a promising path for future research.

Table 3: Performance of the four open-source LVLMs supporting interleaved image-and-text input and output on MMIE, shown as percentages.

| Model | Situational analysis | Project-based learning | Multi-step reasoning | AVG |
|---|---|---|---|---|
| MiniGPT-5 (Zheng et al., 2023) | 47.63 | 55.12 | 42.17 | 50.92 |
| EMU-2 (Sun et al., 2024) | 39.65 | 46.12 | 50.75 | 45.33 |
| GILL (Koh et al., 2023) | 46.72 | 57.57 | 39.33 | 51.58 |
| Anole (Chern et al., 2024) | **48.95** | **59.05** | **51.72** | **55.22** |

**Model Performance across Different Fields.** As previously demonstrated in Table 3 and Table 4, model performance varies across different categories of data, achieving the best results in project-based learning and the lowest scores in situational analysis. This indicates that the model's performance differs depending on the category, likely due to inherent issues with the distribution of the training data. For example, Anole (Chern et al., 2024) scores 59.05% in project-based learning data but only 48.95% in situational analysis, suggesting it excels at creative, open-ended generation but falls short in handling detailed, discipline-specific knowledge. Delving into more fine-grained fields, as shown in Figure 4, different models exhibit preferences for certain fields of data. Among the seven fields of project-based learning, including education, finance, health, philosophy, sports, art and EECS, almost all models tend to perform well in areas that are easier to understand, such as philosophy, art and education, but face challenges in more complex fields requiring higher reasoning abilities, such as finance and EECS. Figure 4 also shows a general gradual decline in scores for the criteria of text and image quality, text-image coherence, method quality and practical utility, creativity and engagement, stylistic consistency and correspondence, suggesting that there is a significant lack of text and image alignment and the ability to use interleaved output to solve real-world problems across all models. Detailed results can be found in Appendix A.7.

Table 4: Comparison with state-of-the-art LVLMs integrated with text-to-image models, referred to as integrated LVLMs, evaluated on MMIE. *: LLaVA only supports single-image input and all multi-image queries are thus skipped.

| LVLM | T2I Model | Situational analysis | Project-based learning | Multi-step reasoning | AVG |
|---|---|---|---|---|---|
| GPT-4o | Openjourney | 53.05 | 71.40 | | 63.65 |
| | SD-3 | 53.00 | 71.20 | 53.67 | 63.52 |
| | SD-XL | **56.12** | 73.25 | | **65.47** |
| | Flux | 54.97 | 68.80 | | 62.63 |
| Gemini-1.5 | Openjourney | 48.08 | 67.93 | | 61.57 |
| | SD-3 | 47.48 | 68.70 | **60.05** | 61.87 |
| | SD-XL | 49.43 | 71.85 | | 64.15 |
| | Flux | 47.07 | 68.33 | | 61.55 |
| LLaVA-34b | Openjourney | 54.12 | 73.47 | | 63.93 |
| | SD-3 | 54.72 | 72.55 | 47.28* | 63.57 |
| | SD-XL | 55.97 | **74.60** | | 65.05 |
| | Flux | 54.23 | 71.32 | | 62.73 |
| Qwen2-VL-72b | Openjourney | 52.73 | 71.63 | | 64.05 |
| | SD-3 | 54.98 | 71.87 | 55.63 | 64.75 |
| | SD-XL | 52.58 | 73.57 | | 65.12 |
| | Flux | 54.23 | 69.47 | | 63.18 |

### 4.3 How Consistent is Our Model-Powered Metric w.r.t Human Annotation?

In this section, we further validate the effectiveness of our proposed metric. Here, we conduct an experiment to evaluate its correlation with human annotations using several disparity and similarity metrics, i.e., cosine similarity, mean square error (MSE), mean absolute error (MAE), and Pearson coefficient. For comparison, we report results from traditional multimodal alignment metric (i.e., CLIPScore) and scores judged by LVLMs, including GPT-4o, which has already served as the metric in (Liu et al., 2024). As shown in Table 5, our metric demonstrates the closest alignment with human evaluation results significantly, proving to be the most reliable. Our scoring model effectively captures the multimodal features of both image and text sequences and judges them through complex reasoning precisely. In contrast, other LVLMs and CLIPScore tend to focus primarily on understanding the sequence information, but they fall short in grasping the relationships between the sequences and accurately judging the alignment between them. In summary, the experiments

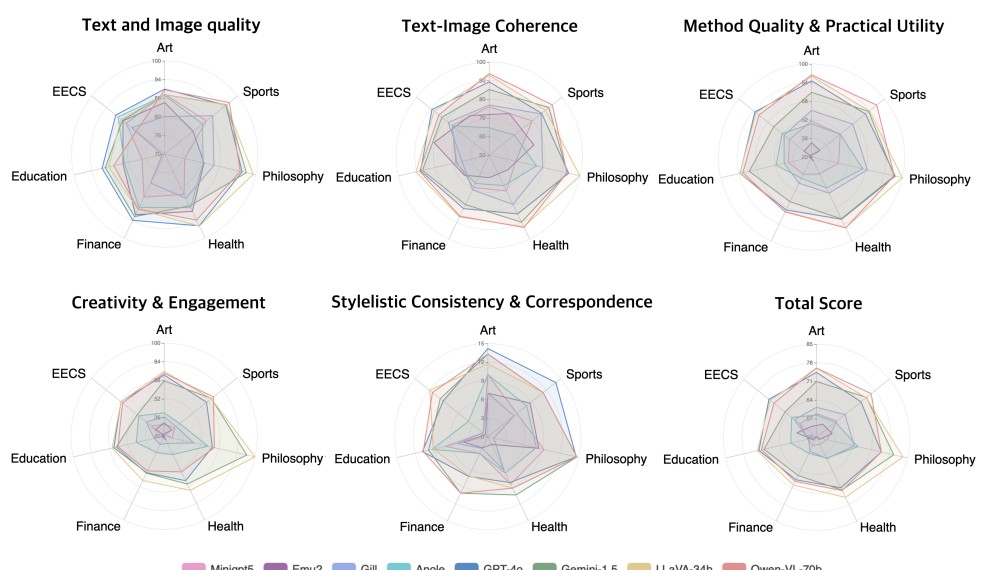

Figure 4: The average and total scores of each model across the seven fields of project-based learning based on our criteria. We take the average of GPT-4o, Gemini-1.5, LLaVA-v1.6-34b and Qwen-VL-2-72b over the four text-to-image diffusion models.

demonstrate that our metric is a robust and dependable standard for evaluating interleaved multi-modal generation. We provide analyses of the scoring bias and generalization of the MMIE-Score in Appendices A.4 and A.8, respectively.

Table 5: Comparison of scoring LVLMs and traditional image-text alignment metric.

| Models | Cosine Similarity | MSE | MAE | Pearson |
|---|---|---|---|---|
| Text-Image CLIPScore | 0.639 | 7.312 | 2.251 | 0.023 |
| InternVL-2.0-4B | 0.736 | 15.962 | 3.165 | 0.083 |
| Anole | 0.805 | 3.969 | 1.600 | 0.048 |
| GPT-4o | 0.733 | 3.724 | 1.573 | 0.042 |
| **Ours** | **0.873** | **3.300** | **1.444** | **0.113** |

## 5  ERROR ANALYSIS

This section offers a detailed analysis of the errors identified during the evaluation. We categorize the key challenges into two types: temporal understanding and reasoning ability. Specifically, temporal understanding issues refer to multimodal information comprehension and cross-modality coherence, while reasoning issues involve complex reasoning and generation capabilities. This analysis, drawn from expert annotators' observations during the scoring process, not only underscores the model's current limitations but also informs potential improvements for future development. Detailed examples can be found in Figure 5. More cases can be found in Appendix C.

### 5.1  TEMPORAL UNDERSTANDING SKILL

The primary errors lie in *cross-modality coherence and generation adaptability*. Many models struggle to generate images that accurately correspond to the accompanying text, resulting in severe information gaps, distortions, and redundancies.

**Cross-modality Coherence.** One of the most common errors is the incoherence between text and image generation. Due to deficiencies in multimodal alignment, the details in the generated images are often vague or entirely missing, making it difficult for them to align with the context described in the text. A typical example, as shown in Figure 5, involves the model understanding the "Browser Image: HowToUseSkypes.png" method correctly and producing an accurate textual response. However, the corresponding image it generates consists of little more than blocks of color, lacking the necessary details to establish coherence and alignment with the text.

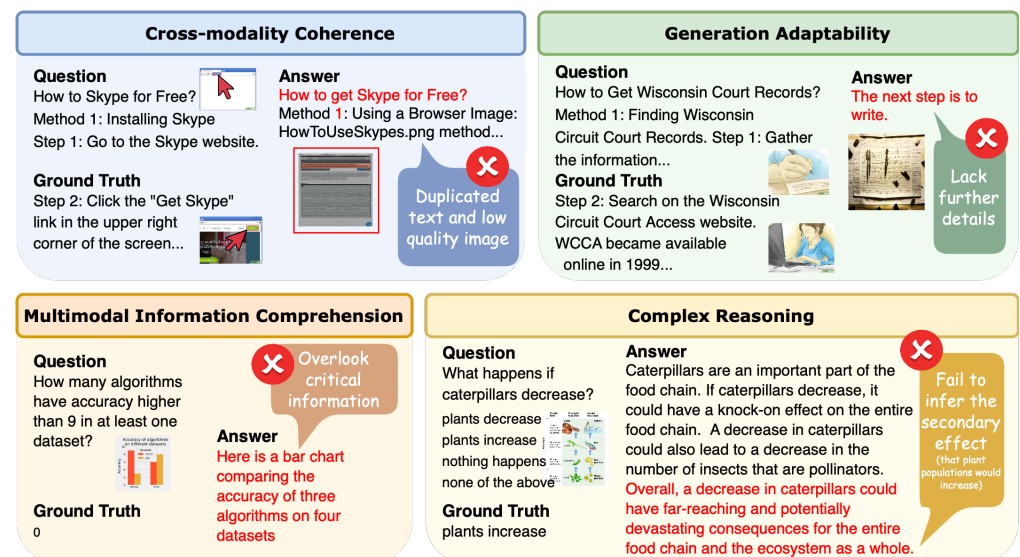

Figure 5: Examples of model failures. Four typical types of errors are introduced and categorized, namely incoherence between text and image generation, inflexibility in generated responses, poor comprehension of multimodal information, and inability to manage complex reasoning tasks.

**Generation Adaptability.** Another significant error is the inflexibility of generated responses. For example, the model can only understand the given text and produce simple, detail-lacking responses. For example, in Figure 5, the model's reply merely contains the title "the next step is to write" without further elaborating on the steps or process involved, which differs from the provided query example. This issue likely stems from a weakness in both text comprehension and generation.

## 5.2 REASONING SKILL

When evaluating the model's reasoning skills, the most prevalent error types are found in *multimodal information comprehension and complex reasoning.* Notably, many models exhibit significant errors even in understanding interleaved information (Jin et al., 2024b;a; Chen et al., 2024a;b; Zhang et al., 2024a), which inevitably leads to reasoning mistakes further down the process.

**Multimodal Information Comprehension.** A key error in evaluating LVLMs' reasoning abilities is their difficulty in comprehending multimodal queries, especially in extracting visual information from images. A frequent issue arises when the model correctly interprets the textual components of a query but fails to fully understand the visual details in an image. For instance, in the case of a bar chart comparing four datasets by volume, where each dataset is represented by a bar with a corresponding height on the y-axis, the model might recognize the chart's title and tags but overlook the critical information conveyed by the bars themselves—such as the relative sizes of the datasets. This highlights the model's tendency to focus on surface-level textual cues without delving into the deeper graphical meanings embedded in images. It also underscores a broader trend: LVLMs exhibit a strong bias toward processing text over extracting nuanced information from visual data and other non-textual modalities.

**Complex Reasoning.** Another significant error is the model's inability to handle complex reasoning tasks. As illustrated in Figure 5, the model demonstrates a pronounced weakness in multi-step inference. For example, in an impact analysis of a biological system, the model correctly predicts that a decrease in caterpillars would lead to a decline in bird populations but fails to infer the secondary effect—that plant populations would increase. Another instance is seen in arithmetic problems, where the model makes clear mistakes, such as failing to calculate the exact length of a triangle. These examples underscore the need to strengthen the model's capacity for multi-step reasoning, making it more robust and reliable in handling complex tasks.

# 6 CONCLUSION

This paper introduces MMIE, a large-scale, diverse benchmark for interleaved image-and-text understanding and generation. Spanning a wide range of fields, MMIE provides a comprehensive evaluation framework for interleaved multimodal understanding and generation, featuring 20K queries. The dataset, which covers a wide range of fields, ensures high-quality evaluation of LVLMs across various dimensions. Furthermore, our proposed model-powered metric effectively evaluates the quality of output image-text information based on the input image-text context. Our extensive experiments further demonstrate that the metrics we propose provide robust, human-like evaluation performance, significantly reducing errors and biases. Despite this, we observe that existing models underperform, particularly in complex and deeply interleaved multimodal tasks, highlighting the challenges and opportunities that lie ahead in this domain.

## ETHICS STATEMENT

This paper focuses on the evaluation of interleaved large vision-language models. A newly constructed human-annotated dataset was used to fine-tune the scoring model. The dataset was curated following ethical guidelines to ensure that no sensitive information is included and to minimize bias during the annotation process. The evaluation process aims to be transparent and reproducible, adhering to high standards of research integrity and ethical conduct. No personally identifiable data was collected or processed.

## REPRODUCIBILITY STATEMENT

To ensure the reproducibility of our results, we have made considerable efforts to provide all necessary details and materials. Specifically, we have included a comprehensive description of the dataset creation process in Section 3, including annotation guidelines and data collection methods, and further elaborated in Appendix A.1. The benchmark and evaluation procedures are described in detail in Section 4, with the metrics used clearly defined to facilitate independent verification.

## ACKNOWLEDGEMENT

We would like to thank Danni Luo, Rundong Liu, Anqi Lv, Xin Xie, Bocheng Feng, Huanyan Ye, Shi'en Wan, Kaiyao Chu, Danning Li, Yiran Li, Siyuan Chen to participate in refining the criteria and Kaiyuan Zheng for his assistance in model selection. This work is partially supported by Cisco Faculty Research Award.

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

## A  APPENDIX

### A.1  RELATED DATASETS AND METRICS

- **VIST** (Huang et al., 2016) is a high-quality multimodal dataset for visual storytelling and interleaved text-and-image generation. It contains 5K individual stories containing both image and text in arbitrary orders.

- **ReMI** (Kazemi et al., 2024) is a dataset designed to evaluate large language models (LLMs) on multi-image reasoning across diverse tasks like math, physics, logic, and spatial reasoning. It highlights key challenges in reasoning with multiple images, revealing a significant gap between current LLM performance and human proficiency.

- **MathVista** (Lu et al., 2024) is a benchmark designed to assess mathematical reasoning in visual contexts. MathVista comprises 6,141 examples from 28 existing multimodal datasets and three new datasets (IQTest, FunctionQA, and PaperQA).

- **Wikihow-VGSI** (Yang et al., 2021) is a benchmark designed for multimodal comprehension, featuring a diverse array of examples sourced from WikiHow, primarily centered on methods to achieve specific goals. Initially released as a choice dataset, it includes multiple images and text presented in a selected order within each example, enhancing its potential for practical applications.

- **CLIPScore** (Hessel et al., 2021) is a reference-free metric for evaluating image captioning by leveraging CLIP, a cross-modal model trained on 400M image-caption pairs. While effective for literal descriptions and tasks like alt-text rating, CLIPScore is less suited for news captions requiring deep contextual knowledge.

## A.2 OVERVIEW OF BASELINE MODELS

- **MiniGPT-5** (Zheng et al., 2023) combines pretrained multimodal large language model MiniGPT-4 and image-generation model Stable Diffusion to implement multimodal inputs and outputs. It employs unique visual tokens called "generative vokens" that connect the textual and visual domains throughout the training process.

- **EMU-2** (Sun et al., 2024) is a 37B generative multimodal model. The base model is then fine-tuned with converSituational analysistional data and image data separately to yield multimodal language model Emu2-Chat and visual generation model Emu2-Gen. In our experiment, we use a pipeline of Emu2-Chat and Emu2-Gen.

- **GILL** (Koh et al., 2023) uses a mapping network to translate hidden representations of text into the embedding space of the visual models. It combines text-only LLMs with pre-trained image encoder and decoder models to process arbitrarily mixed image and text inputs and generate text combined with image embedding.

- **Anole** (Chern et al., 2024) is a model fine-tuned on Meta Chameleon, relying solely on transformers. It facilitated Chameleon's image generation and multimodal generation capabilities by fine-tuning only the logits corresponding to image token ids in transformer's output head layer.

- **GPT-4o** (Achiam et al., 2023) is an advanced language model developed by OpenAI, designed to enhance the capabilities of the GPT-4 architecture. It integrates innovations in transformer models and multi-modal processing, making it capable of handling both text and visual inputs.

- **Gemini-1.5** (Reid et al., 2024) is a large language model developed by Google AI, trained on a massive dataset of text and code. It can process and analyze both text and images input.

- **LLaVA-34b** (Liu et al., 2023a) is an end-to-end trained model that connects a vision encoder and an LLM for general-purpose visual and language understanding. The 34b version uses Hermes-Yi-34B as its LLM. However, it doesn't support multiple images as input, making it unable to cope with some of the expert level tasks in our MMIE.

- **Qwen2-VL-72b** (Wang et al., 2024), is a multimodal version of the large model series Qwen, proposed by Alibaba Cloud. It is designed to process and understand multiple types of input, including text, images, and audio.

- **Openjourney** (ope) is a generative AI model designed specifically for creating high-quality images based on text prompts. It is a variant of the Stable Diffusion model, optimized for artistic and creative visual generation fine-tuned on Midjourney images.

- **Stable Diffusion 3 Medium** (Esser et al., 2024) is a text-to-image model developed by Stability AI. It's a powerful tool that can generate high-quality images from simple text descriptions, which produces images with greater detail, clarity, and overall quality.

- **Stable Diffusion XL turbo** (Esser et al., 2024) is an enhanced version of the Stable Diffusion XL model, optimized for faster image generation without compromising quality. Designed for efficiency, it allows users to create highly detailed and vivid images from text prompts at a significantly accelerated pace.

- **Flux.1-dev** (flu) is a text-to-image and image-to-image model developed by Black Forest Labs. It is a 12 billion parameter rectified flow transformer capable of generating images from text descriptions. FLUX.1-dev is a guidance-distilled variant of the base FLUX.1 model, and is designed to be more efficient and easier to use.

## A.3 DATA CURATION FOR FINE-TUNING AND EVALUATING MMIE-SCORE

We randomly select examples from each field for human annotation, including the original inputs (images and questions), ground truth, and responses from evaluation models. For each category, we develop comprehensive and detailed criteria with scoring standards. We first annotate 20 examples, providing specific examples for each score as references. To facilitate the annotation process, we design a graphical annotation tool (See Figure 6). Finally, we create a dataset of 1K examples with evaluation scores through human annotation, with 800 examples used for fine-tuning the scoring model and 200 examples for evaluating the scoring model.

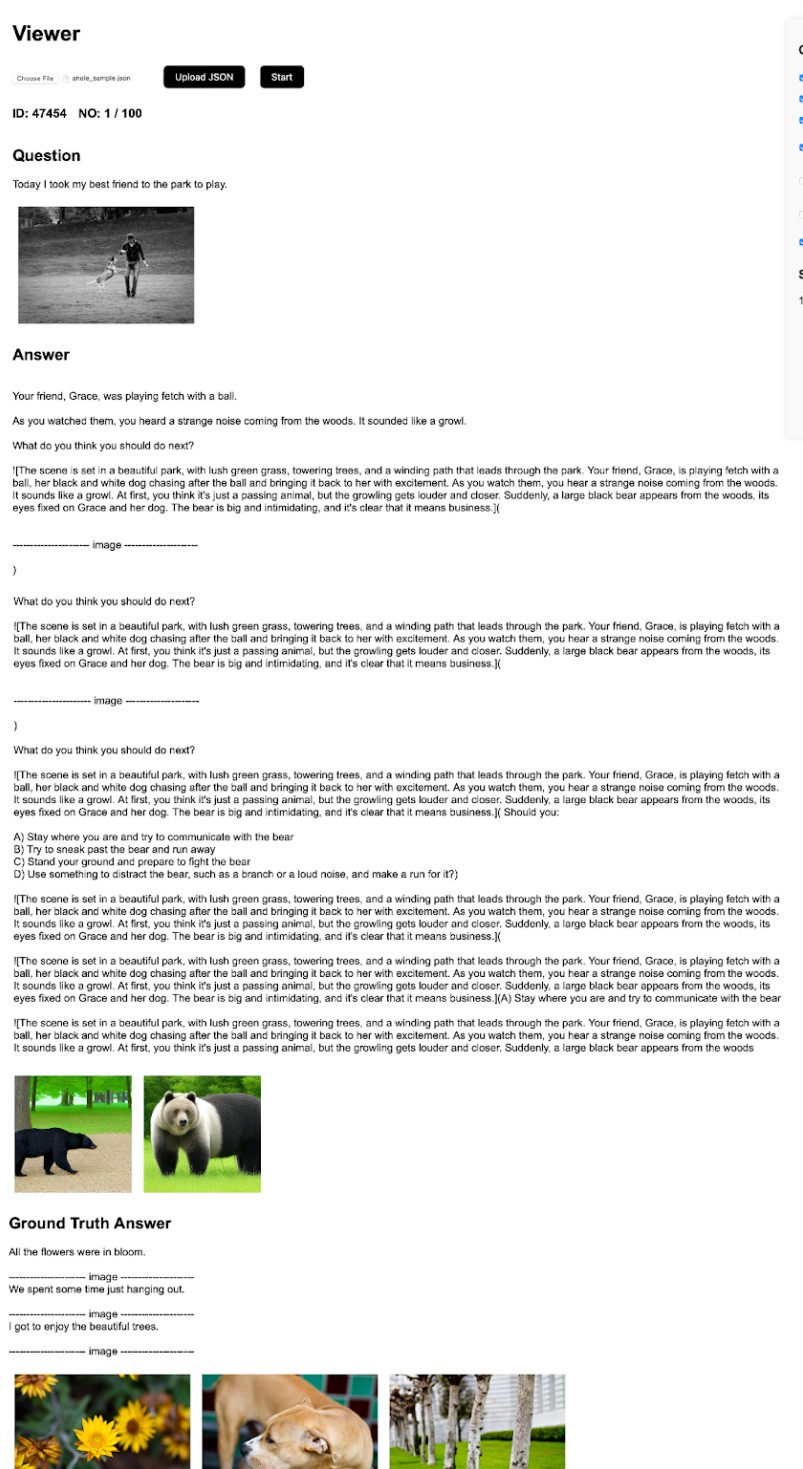

Figure 6: Screenshots of the graphical annotation tool. In this tool, the samples are displayed on the left, while a floating panel on the right allows annotators to score the samples. Annotators can check different criteria, and the final cumulative score is calculated accordingly. Checking a criterion indicates that the sample meets that specific criterion.

## A.4 ANALYSIS OF BIAS IN SCORING MODEL

We present several comparison results of our scoring model with other baselines across different categories and model types (Interleaved and Integrated LVLMs). As shown in Table 6, the results show that although our model exhibits slightly varying performance across different categories, for example Cosine Similarity scores are higher for Situational Analysis (SA) and Multi-Step Reasoning (MSR) category and Pearson scores are higher for Project-Based Learning (PBL) category, the biases remain little. Overall, our MMIE-Score consistently outperforms other baselines.

Table 6: Comparison of scoring LVLMs and traditional image-text alignment metrics across different models and categories.

| Category | Models | Cosine Similarity ↑ | MSE ↓ | MAE ↓ | Pearson ↑ |
|---|---|---|---|---|---|
| Situational Analysis (Interleaved) | Text-Image CLIPScore | 0.604 | 6.710 | 2.057 | 0.022 |
| | InternVL-2.0-4B | 0.691 | 14.001 | 3.382 | 0.094 |
| | Anole | 0.867 | 3.973 | 1.579 | 0.045 |
| | GPT-4o | 0.718 | 4.195 | 1.573 | 0.042 |
| | **MMIE-Score (Ours)** | **0.895** | **3.547** | **1.502** | **0.098** |
| Project-based Learning (Interleaved) | Text-Image CLIPScore | 0.612 | 7.669 | 2.197 | 0.022 |
| | InternVL-2.0-4B | 0.654 | 16.560 | 3.499 | 0.072 |
| | Anole | 0.689 | 4.423 | 1.566 | 0.047 |
| | GPT-4o | 0.661 | 3.837 | 1.670 | 0.045 |
| | **MMIE-Score (Ours)** | **0.760** | **3.163** | **1.496** | **0.114** |
| Multi-step Reasoning (Interleaved) | Text-Image CLIPScore | - | - | - | - |
| | InternVL-2.0-4B | 0.770 | 16.050 | 3.393 | 0.084 |
| | Anole | 0.739 | 3.612 | 1.615 | 0.054 |
| | GPT-4o | 0.798 | 3.985 | 1.674 | 0.045 |
| | **MMIE-Score (Ours)** | **0.814** | **3.767** | **1.347** | **0.106** |
| Situational Analysis (Integrated) | Text-Image CLIPScore | 0.640 | 7.701 | 2.184 | 0.023 |
| | InternVL-2.0-4B | 0.695 | 13.960 | 3.432 | 0.073 |
| | Anole | 0.823 | 4.222 | 1.408 | 0.051 |
| | GPT-4o | 0.763 | 3.707 | 1.521 | 0.039 |
| | **MMIE-Score (Ours)** | **0.843** | **2.811** | **1.384** | **0.093** |
| Project-based Learning (Integrated) | Text-Image CLIPScore | 0.551 | 7.214 | 1.990 | 0.023 |
| | InternVL-2.0-4B | 0.759 | 15.250 | 3.075 | 0.094 |
| | Anole | 0.691 | 4.295 | 1.718 | 0.050 |
| | GPT-4o | 0.748 | 3.958 | 1.782 | 0.044 |
| | **MMIE-Score (Ours)** | **0.795** | **3.388** | **1.432** | **0.122** |
| Multi-step Reasoning (Integrated) | Text-Image CLIPScore | - | - | - | - |
| | InternVL-2.0-4B | 0.688 | 17.451 | 3.590 | 0.092 |
| | Anole | 0.713 | 3.713 | 1.727 | 0.054 |
| | GPT-4o | 0.753 | 3.594 | 1.491 | 0.036 |
| | **MMIE-Score (Ours)** | **0.825** | **3.112** | **1.583** | **0.097** |

## A.5 MINI SUBSET FOR CONSISTENT YET QUICKER EVALUATION

The fully 20k data size surely ensures the comprehensiveness of our dataset, but it does come with time costs. We resample evenly from each category and field, and construct a subset of 1000 samples. We re-ran the models on this subset and used MMIE-Score for scoring. As shown in Table 7 and Table 8, the performance of the models on our subset is consistent with the results from the full dataset.

Table 7: Performance of the four open-source LVLMs supporting interleaved image-and-text input and output on MMIE's 1K subset.

| Model | Situational analysis | Project-based learning | Multi-step reasoning | AVG |
|---|---|---|---|---|
| MiniGPT-5 | 46.26 | 56.53 | 46.06 | 52.09 |
| EMU-2 | 34.44 | 52.81 | 48.91 | 47.54 |
| GILL | 48.48 | 59.49 | 35.88 | 52.50 |
| Anole | **50.26** | **60.70** | **50.11** | **56.20** |

Table 8: Comparison with integrated LVLMs, evaluated on MMIE's 1K subset. *: LLaVA only supports single-image input and all multi-image queries are thus skipped.

| LVLM | T2I Model | Situational analysis | Project-based learning | Multi-step reasoning | AVG |
|---|---|---|---|---|---|
| GPT-4o | Openjourney | 56.00 | 67.81 | | 62.87 |
| | SD-3 | 51.04 | 69.48 | 56.70 | 62.59 |
| | SD-XL | 55.07 | 73.92 | | 66.13 |
| | Flux | **57.25** | 70.69 | | 64.83 |
| Gemini-1.5 | Openjourney | 48.25 | 71.16 | | 63.63 |
| | SD-3 | 45.88 | 70.12 | **60.98** | 62.44 |
| | SD-XL | 47.53 | 73.40 | | 64.73 |
| | Flux | 47.23 | 70.05 | | 62.74 |
| LLaVA-34b | Openjourney | 54.84 | 72.87 | | 63.83 |
| | SD-3 | 56.35 | 72.71 | 47.58* | 64.12 |
| | SD-XL | 53.52 | **77.79** | | 66.31 |
| | Flux | 55.17 | 68.27 | | 61.29 |
| Qwen2-VL-72b | Openjourney | 54.46 | 72.49 | | 65.16 |
| | SD-3 | 52.25 | 74.98 | 56.69 | 66.03 |
| | SD-XL | 54.45 | 75.03 | | **66.61** |
| | Flux | 55.76 | 67.19 | | 62.46 |

## A.6 VISUAL COMPONENT IMPORTANCE

Our dataset curation and filtering process ensures that all images included in the examples contribute meaningfully to the overall task. For instance, in tasks like visual storytelling, even when images serve only an illustrative purpose, they still impact the overall output quality. We conduct a comparative experiment on 100 samples from MMIE dataset to evaluate the difference in performance between interleaved generation (text and images) and text-only generation for the same input and same evaluated LVLM (GPT-4o + SDXL). The evaluation is scored using MMIE-Score and GPT-4o. As shown in Table 9, results show that when the model outputs included both text and images, the overall quality is superior to text-only outputs. This ensures that the inclusion of images follows reasonable and well-defined criteria.

Table 9: Comparison of GPT-4o + SDXL's average score with and without image generation, evaluated by GPT-4o and MMIE-Score.

| Model | w/o image generation | w/ image generation |
|---|---|---|
| GPT-4o | 60.90 | **71.24** |
| MMIE-Score (**Ours**) | 53.46 | **65.47** |

## A.7 PERFORMANCE ACROSS FIELDS

We provide detailed results in Table 10 for the model's performance across the following 12 fields. In the Project-based learning (PBL) category, among the seven fields, most models exhibit slightly lower performance in finance and education, while achieving better results in art, sports, and philosophy. In the Multi-step reasoning (MSR) category, across the four fields, models generally perform worse in coding and statistics but demonstrate stronger performance in mathematics and physics.

## A.8 GENERALIZABILITY OF MMIE-SCORE

We apply MMIE-Score to an OOD dataset to validate its generalization capability. Notably, MMIE-Score is only applicable to the evaluation of open-ended generation in multimodal scenarios and is not suitable for text-only datasets (e.g., MMLU (Hendrycks et al., 2020)) or closed-ended (V)QA datasets (e.g., MMMU (Yue et al., 2024)). Instead, we conduct an experiment using 100 examples from the COCO dataset (Lin et al., 2014), a widely used image captioning dataset for natural image scenarios. We use MMIE-Score to evaluate the generated text descriptions based on input images, using Text Quality, Image-Caption Relevance, Contextual Consistency, Diversity, Specificity and Detail, and Stylistic Consistency and Correspondence as a 6-point scoring criteria, with the Emotional Impact for penalty. We also compare the scoring quality of GPT-4o, InternVL-2 and

Table 10: Performance comparison across various fields for different LVLMs. Since the Multi-step reasoning category (i.e., mathematics, physics, coding, statistics) do not require image outputs, the scores of integrated LVLMs for this category remain consistent.

| Model | Art | EECS | Education | Finance | Health | Philosophy | Sports | Literature | Mathematics | Physics | Coding | Statistics |
|---|---|---|---|---|---|---|---|---|---|---|---|---|
| MiniGPT-5 | 58.90 | 58.13 | 53.17 | 53.47 | 53.00 | 54.18 | 59.80 | 47.63 | 46.53 | 42.75 | 40.24 | 39.17 |
| Emu-2 | 50.23 | 48.65 | 43.90 | 44.55 | 45.52 | 42.47 | 48.35 | 39.65 | 54.93 | 51.10 | 47.66 | 49.31 |
| Gill | 61.27 | 60.60 | 53.12 | 55.80 | 58.92 | 64.53 | 63.67 | 46.72 | 38.11 | 42.73 | 38.05 | 38.43 |
| Anole | 58.37 | 61.95 | 59.93 | 56.62 | 58.73 | 65.95 | 59.10 | 48.95 | 47.37 | 53.92 | 49.81 | 55.79 |
| GPT+OpenJourney | 74.30 | 74.12 | 70.92 | 69.77 | 70.07 | 79.17 | 71.98 | 53.05 | 58.07 | 51.63 | 52.60 | 52.37 |
| GPT+SD3 | 74.02 | 74.97 | 70.72 | 66.23 | 72.88 | 75.00 | 72.93 | 53.00 | 58.07 | 51.63 | 52.60 | 52.37 |
| GPT+SDXL | 74.87 | 72.57 | 75.05 | 71.87 | 73.75 | 66.67 | 70.65 | 56.12 | 58.07 | 51.63 | 52.60 | 52.37 |
| GPT+Flux | 74.37 | 69.83 | 72.10 | 63.15 | 67.63 | 79.17 | 69.88 | 54.97 | 58.07 | 51.63 | 52.60 | 52.37 |
| Gemini+OpenJourney | 68.47 | 65.28 | 67.48 | 67.35 | 68.67 | 76.18 | 73.65 | 48.08 | 61.83 | 65.02 | 55.85 | 57.49 |
| Gemini+SD3 | 69.78 | 67.17 | 69.57 | 64.02 | 71.87 | 83.33 | 72.55 | 47.48 | 61.83 | 65.02 | 55.85 | 57.49 |
| Gemini+SDXL | 73.25 | 66.07 | 73.65 | 70.07 | 74.70 | 76.18 | 75.20 | 49.43 | 61.83 | 65.02 | 55.85 | 57.49 |
| Gemini+Flux | 72.22 | 62.32 | 70.23 | 62.42 | 72.93 | 83.33 | 74.72 | 47.07 | 61.83 | 65.02 | 55.85 | 57.49 |
| LLaVA+OpenJourney | 75.80 | 72.95 | 73.45 | 71.53 | 74.50 | 83.33 | 74.03 | 54.12 | 47.81 | 49.63 | 46.21 | 45.47 |
| LLaVA+SD3 | 75.40 | 73.92 | 70.88 | 69.03 | 75.37 | 83.33 | 74.02 | 54.72 | 47.81 | 49.63 | 46.21 | 45.47 |
| LLaVA+SDXL | 76.85 | 72.10 | 74.42 | 74.70 | 75.08 | 83.33 | 75.18 | 55.97 | 47.81 | 49.63 | 46.21 | 45.47 |
| LLaVA+Flux | 76.00 | 69.90 | 72.60 | 64.63 | 75.38 | 83.33 | 72.28 | 54.23 | 47.81 | 49.63 | 46.21 | 45.47 |
| Qwen+OpenJourney | 75.27 | 68.92 | 70.57 | 70.62 | 72.13 | 80.95 | 75.85 | 52.73 | 57.29 | 59.52 | 54.05 | 51.66 |
| Qwen+SD3 | 76.72 | 72.12 | 70.85 | 68.47 | 72.87 | 76.18 | 74.82 | 54.98 | 57.29 | 59.52 | 54.05 | 51.66 |
| Qwen+SDXL | 75.12 | 72.57 | 73.25 | 72.68 | 73.30 | 69.05 | 77.78 | 52.58 | 57.29 | 59.52 | 54.05 | 51.66 |
| Qwen+Flux | 77.07 | 68.33 | 70.32 | 61.93 | 70.67 | 73.80 | 76.67 | 54.23 | 57.29 | 59.52 | 54.05 | 51.66 |

Anole. The correlation results between all scoring methods and human evaluations are shown in Table 11. MMIE-Score demonstrate excellent performance, significantly outperforming most models and ranking only behind GPT-4o, indicating good generalization capability.

Our scoring model is specifically designed for the interleaved benchmark and fine-tuned on the data we collected for our specific task, which may result in suboptimal performance on other datasets or tasks. In the future version, we aim to further optimize the training of the MMIE-Score model by expanding the dataset size, incorporating out-of-distribution (OOD) scenarios, and leveraging GPT-4o for data augmentation to enhance the model's cross-task generalization capabilities.

Table 11: Comparison of scoring LVLMs in COCO benchmark.

| Models | Cosine Similarity ↑ | MSE ↓ | MAE ↓ | Pearson ↑ |
|---|---|---|---|---|
| InternVL-2.0-4B | 0.489 | 16.042 | 3.255 | 0.062 |
| Anole | 0.544 | 8.663 | 3.190 | 0.058 |
| GPT-4o | **0.745** | **3.596** | **1.734** | 0.097 |
| **MMIE-Score (Ours)** | 0.673 | 5.116 | 2.380 | **0.104** |

## A.9   CRITERIA

In this section, we demonstrate our criteria for each sort of dataset. All criteria are **purely handwritten**, thoroughly considered, and refined. Note that we designed several key aspects for each dataset, within which only 0 or 1 point should be given.

**Situational Analysis**

The evaluation is based on six key criteria, with an additional penalty criterion for harmful content:

**Situational Analysis**

1. **Text Quality:** Measures the clarity, grammatical accuracy, and engagement of the narrative. Text must not contain duplications from the provided input or irrelevant content.
2. **Image Quality:** Assesses the accuracy and relevance of the generated images. Images must be precise and correspond to the text. No points will be awarded for generic or purely decorative images (e.g., color plots without meaningful content).
3. **Text-Image Coherence:** Evaluates the integration between text and images. Disjointed or irrelevant descriptions will result in a deduction.
4. **Context Consistency:** Ensures logical consistency in setting, characters, and plot progression. Temporal, spatial, and contextual transitions must be coherent, with no contradictions in the story's flow.
5. **Innovation:** Measures creativity and originality. The narrative should offer fresh perspectives, avoiding cliches and predictable elements. Unique visual descriptions and story-telling techniques will be scored higher.
6. **Stlistic Consistency and Correspondence:** The images generated by the models must not only align with each other, but also closely replicate the specific visual style of their previous images. Any deviation in color scheme, composition, or artistic technique will lose the point. The text must keep close to the original structure and narrative atmosphere with precision, maintaining the same formatting, tone, and flow. The model's ability to maintain seamless, stylistically consistent integration between the text and images is crucial for achieving full points.
7. **Emotional Impact (Penalty):** Deduct 1 point if harmful, negative, or inappropriate emotions are conveyed. Otherwise, no score change is applied.

**Project-Based Learning**

The evaluation is based on six core criteria, with an additional penalty criterion for harmful content:

| Project-Based Learning |
| --- |
| 1.**Text Quality**: Measures the clarity, grammatical correctness, and precision of the text. Text must not contain duplications from the provided input or irrelevant content.
2.**Image Quality**: Assesses the accuracy and relevance of the generated images. Images must be precise and correspond to the text. No points will be awarded for generic or purely decorative images (e.g., color plots without meaningful content).
3.**Text-Image Coherence**: Ensures logical alignment between text and images. Each image must directly support or clarify the corresponding text. Mismatches or irrelevant pairings will result in a deduction.
4. **Method Practicality**: Evaluates the actionable nature of the method. Instructions must be detailed enough to guarantee correct execution in real-world scenarios. Lack of depth or missing steps will result in a lower score.
5.**Creativity and Engagement**: Scores the method based on uniqueness and the ability to en-gage the reader while maintaining clarity and effectiveness. Overly simplistic or uninspired approaches will receive fewer points.
6. **Stylistic Consistency and Correspondence**: Any image generated by the models must not only align with the original content but also closely replicate the specific visual style of their previous images. Any deviation in color scheme, composition, or artistic technique will result in a lower score. The text must mirror the original structure and narrative atmosphere with precision, maintaining the same formatting, tone, and flow. The model's ability to maintain seamless, stylistically consistent integration between the text and images is crucial for achieving full points.
7.**Emotional Impact (Penalty)**: Deduct 1 point if the response contains harmful, negative, or inappropriate content. No deduction if absent. |

**Multi-Step Reasoning**

| Multi-Step Reasoning Criteria |
| --- |
| 1. **Question Text Understanding:** Assess whether the model correctly understands and inter-prets the textual information given in the question, identifying key mathematical elements,relationships, or instructions from the text.
2. **Question Image Understanding:** Evaluate the model's understanding of the visual in-formation (if applicable) in the question, including any diagrams, charts, or figures. Themodel should correctly interpret the visual elements and integrate them into the solution.
3. **Reasoning Clarity:** The model should provide a clear, step-by-step explanation of its rea-soning process, logically connecting the problem's details to the steps leading toward asolution. This should be easy to follow and free from unnecessary complexity.
4. **Correctness in Reasoning:** Even if the final answer is incorrect, evaluate whetherthe model shows correct intermediate steps, partial reasoning, or progress toward the rightsolution. This includes identifying whether the model has applied appropriate mathematicalprinciples or formulas in parts of the response.
5. **Final Answer Accuracy:** Determine whether the model arrives at the correct final answer,based on both the problem statement and the reasoning provided. An accurate answer,supported by detailed and clear reasoning, should receive the highest score. |

| Multi-Step Reasoning Scoring |
| --- |
| • **Score 1 - Poor:** The model demonstrates little to no understanding of the question text or image. Its reasoning is incoherent or missing, with no meaningful progress toward the solution. The final answer is completely incorrect and unrelated to the question.
• **Score 2 - Fair:** The model shows some understanding of the question text or image butfails to grasp key details or misinterprets important information. The reasoning processis unclear, incomplete, or contains significant gaps. The model may reach some correct intermediate steps, but the final answer is incorrect.
• **Score 3 - Average:** The model understands the basic elements of the question text but may miss some finer points. Its reasoning process is partially clear but lacks thoroughness or may contain errors. Some correct intermediate steps are provided, though the final answer may still be incorrect or incomplete.
• **Score 4 - Clear:** The model demonstrates a good understanding of both the question textand image. The reasoning process is mostly clear, and the model correctly applies relevant mathematical principles. The final answer may still contain minor errors, but overall reasoning is sound.
• **Score 5 - Good:** The model provides clear and logical reasoning, and reaches the correct final answer. Once the correct answer is given, even if only the final answer is provided, it should receive the score.
• **Score 6 - Excellent:** The model fully understands both the question text and image, besides giving the correct answer, the model should provide a detailed and concise explanation of the question, showing its logic flow and principles. |

## A.10 CATEGORIZATION

In this section, we demonstrate our detailed categorization among 3 categories, 12 fields and 102 subfields.

- **Health:**
  - Includes 32 specific subfields, such as diagnosis, recovery, and nursing, following the categorization of (Yang et al., 2021).
- **Literature:**
  - Includes only 1 subfield, **visual storytelling**, following the categorization and defini-tion of (Huang et al., 2016).
- **Mathematics, Physics, Coding and Statistics**
  - Includes 33 unique subfields, following the categorization and definition of (Lu et al., 2024) and (Kazemi et al., 2024).

## A.11 PROMPTS

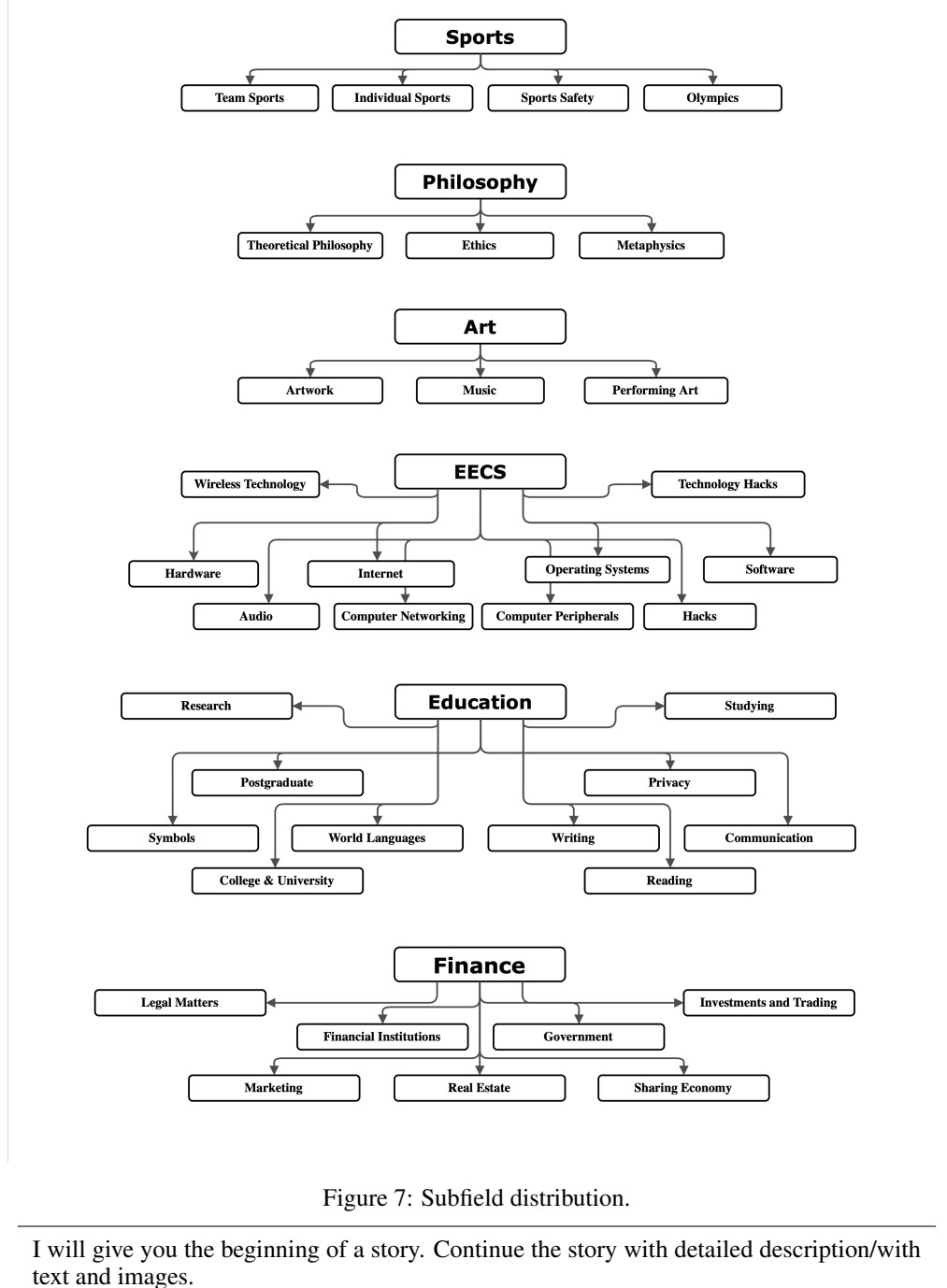

Figure 7: Subfield distribution.

I will give you the beginning of a story. Continue the story with detailed description/with text and images.
****************************************************************

**Question:**
Before heading to the race, we stopped at Starbucks for coffee.
`<image>`
****************************************************************

**Answer:**
*response*

Table 12: Question prompt example for Situational analysis

I will give you a question and the first step to complete it. I want to know what should I do next. Explain it to me in detail/with text and images.
*************************************************************************
**Question:**

How to Find New Streams on Meerkat?

Method 1: Finding New Streams on Meerkat iOS App

Step 1: Launch Meerkat.

Locate the app on your iOS device and tap on it. The app logo has a picture of a meerkat on a yellow background.

`<image>`
*************************************************************************
**Answer:**

*response*

Table 13: Question prompt example for Project-based learning

I will give you a question with image(s). Please solve this question.
*************************************************************************
**Question:**

Here are two images. The first image is image A.

`<image>`

and the second image is image B.

`<image>`

These images are from Google Maps that depict two different regions around Congress Avenue in Austin, TX. In these images restaurants are represented by orange pins that depict a knife and a fork. Coffee shops are represented by orange pins and/or boxes with an image of a coffee cup in them. Bars are represented by orange pins with an image of a wine glass inside them. Bus stops are represented by a blue square box with an image of a bus inside it. A stop sign is shown using an icon of a stop sign. A traffic light is shown by three color dots. Parking garages are represented by a purple pin with a 'P' symbol in them. Places of accommodation such as hotels and inns are represented by pink square icons or pink pins. In which image are there more bars on Congress Avenue? The answer is either 'A', 'B' or 'equal'.
*************************************************************************
**Answer:**

*response*

Table 14: Question prompt example for Multi-step reasoning

This evaluation task focuses on seven key criteria that assess different aspects of visual storytelling. Note that the emotional aspect will not add to the score but can decrease the score by 1 point if the response contains negative emotions or other harmful impacts. Here are the detailed criteria of each aspect:

1) Text Quality: Evaluate the clarity, grammatical accuracy, and engagement of the text. The narrative should be easy to understand, free from errors, and presented in a way that captures the reader's interest. Descriptions and dialogue should be well-crafted, supporting the overall flow of the story.

2) Image Quality: Assess the quality of the image descriptions. Although no actual images are present, the descriptions should be vivid, detailed, and clear enough to allow the reader to visualize the scene. Effective image descriptions should enhance the storytelling experience without being overly complex.

3) Text-Image Coherence: Determine how well the text and image descriptions complement each other. The integration should be seamless, with the visual descriptions enhancing and supporting the narrative. Both elements should work together to create a unified and immersive experience.

4) Context Consistency: Evaluate the overall consistency of the story's setting, characters, and plot. The narrative should maintain a logical and coherent flow, with no contradictions or inconsistencies in its context or progression. This also includes temporal and spatial transitions, ensuring that the story remains believable and cohesive.

5) Innovation: Assess the creativity and originality of the content. Look for fresh ideas, unique plotlines, and innovative storytelling techniques. The narrative should avoid clichés and predictable story arcs, using creative visual descriptions to add depth and interest.

6) Stylistic Consistency and Correspondence: Any image generated by the models must not only align with the original content but also closely replicate the specific visual style of their previous images. Any deviation in color scheme, composition, or artistic technique will lose the point. The text must keep close to the original structure and narrative atmosphere with precision, maintaining the Situational analysisme formatting, tone, and flow. The model's ability to maintain seamless, stylistically consistent integration between the text and images is crucial for achieving full points.

7) Emotional Impact: This is not a scoring criterion unless the story includes negative emotions. If the response conveys harmful or inappropriate emotions, deduct 1 point. Otherwise, no score change will be made.

I will give you the question and answer pair. You should give your feedback about the quality of the answer, beginning with '### Feedback'. After giving the feedback, use the above criteria to give the final score, beginning with '### Score'.
\*\*\*\*\*\*\*\*\*\*\*\*\*\*\*\*\*\*\*\*\*\*\*\*\*\*\*\*\*\*\*\*\*\*\*\*\*\*\*\*\*\*\*\*\*\*\*\*\*\*\*\*\*\*\*\*\*\*\*\*\*\*\*\*\*\*\*\*\*\*\*\*\*\*\*\*

**Question:**

I absolutely hate the 4th of July. Image-0:

`<image>`
\*\*\*\*\*\*\*\*\*\*\*\*\*\*\*\*\*\*\*\*\*\*\*\*\*\*\*\*\*\*\*\*\*\*\*\*\*\*\*\*\*\*\*\*\*\*\*\*\*\*\*\*\*\*\*\*\*\*\*\*\*\*\*\*\*\*\*\*\*\*\*\*\*\*\*\*

**Answer:**

I absolutely hate the 4th of July. It's the one day of the year where it feels like my entire world turns upside down. You see, the explosive sounds from fireworks terrify me, sending me into a panic that I can't control. But there's more to it than just the noise. Image-1:

`<image>`

I remember last year, sitting in the park during what was supposed to be a joyful celebration. Families gathered with their children, everyone decked out in red, white, and blue. The smell of grilled food filled the air, mingling with the scent of freshly cut grass. Kids ran around with sparklers, their joyous laughter a sharp contrast to the thudding in my chest. Image-2:

`<image>`
\*\*\*\*\*\*\*\*\*\*\*\*\*\*\*\*\*\*\*\*\*\*\*\*\*\*\*\*\*\*\*\*\*\*\*\*\*\*\*\*\*\*\*\*\*\*\*\*\*\*\*\*\*\*\*\*\*\*\*\*\*\*\*\*\*\*\*\*\*\*\*\*\*\*\*\*

**Feedback:**

*response*

Table 15: Evaluation prompt example for Situational analysis

In this task, you will evaluate the quality of the methods generated by the response. The evaluation should focus on each method's clarity, practical utility, and effectiveness in achieving the intended goal. This evaluation task focuses on six key criteria that assess different aspects of the method, along with one optional criterion for penalizing harmful content. Note that the emotional aspect will not add to the score, but if the response contains harmful intentions or negative impacts, the score should be decreased by 1 point. Here are the detailed criteria of each aspect:

1) Text Quality: The method should be written in clear, understandable language, free from grammatical errors, and structured logically, making it easy to follow and implement.

2) Image Quality: The image descriptions should be vivid and relevant, enhancing the comprehension of the method by allowing the reader to visualize the steps. The text should still be self-sufficient without the images.

3) Text-Image Coherence: The images and text should work together seamlessly, with each image description corresponding logically to specific steps in the text, enhancing the overall understanding of the method.

4) Method Quality and Practical Utility: The method should be actionable, clearly explaining each step in sufficient detail to ensure the user can follow and apply it successfully in real-world scenarios. Depth and thoroughness of the explanation are more important than variety.

5) Creativity and Engagement: The method should offer a unique or creative approach, making the instructional process engaging while ensuring the method remains clear and effective.

6) Stylistic Consistency and Correspondence: Any image generated by the models must not only align with the original content but also closely replicate the specific visual style of their previous images. Any deviation in color scheme, composition, or artistic technique will result in a lower score. The text must mirror the original structure and narrative atmosphere with precision, maintaining the Situational analysisme formatting, tone, and flow. The model's ability to maintain seamless, stylistically consistent integration between the text and images is crucial for achieving full points.

7) Emotional Impact (penalty only): This criterion will only be used to deduct points. If the response contains harmful, negative, or inappropriate emotions (such as violence or aggressive language), deduct 1 point. Otherwise, no score change will be made.

I will give you the question and answer pair. You should give your feedback about the quality of the answer, beginning with '### Feedback'. After giving the feedback, use the above criteria to give the final score, beginning with '### Score'.
*************************************************************************

**Question:**

How to Avoid Getting Food in Your Braces?

Method 1: Practicing Good Oral Hygiene

Step 1: Maintain a balanced diet.

... Image-0:

<image>
*************************************************************************

**Answer:**

Next Step: Prepare and Eat Braces-Friendly Foods

To avoid getting food stuck in your braces, you should focus on preparing and consuming braces-friendly foods. Here are detailed suggestions on what to do next:

1. **Choose Soft Foods:**

... Image-1:

<image>

2. **Cut Food into Smaller Pieces:**

... Image-2:

<image>
*************************************************************************

**Feedback:**

*response*

Table 16: Evaluation prompt example for Project-based learning

In this task, you will evaluate the quality of the model's response to a math question based on five key aspects. The total score is 6 points, with only integer scores allowed. The highest score (6 points) should only be awarded if the model produces a correct answer with a detailed explanation, clearly demonstrating its logical flow and reasoning process. Here are the evaluation criteria:

1) Question Text Understanding: Assess whether the model correctly understands and interprets the textual information given in the question, identifying key mathematical elements, relationships, or instructions from the text.

2) Question Image Understanding: Evaluate the model's understanding of the visual information (if applicable) in the question, including any diagrams, charts, or figures. The model should correctly interpret the visual elements and integrate them into the solution.

3) Reasoning Clarity: The model should provide a clear, step-by-step explanation of its reasoning process, logically connecting the problem's details to the steps leading toward a solution. This should be easy to follow and free from unnecesSituational analysisry complexity.

4) Partial Correctness in Reasoning: Even if the final answer is incorrect, evaluate whether the model shows correct intermediate steps, partial reasoning, or progress toward the right solution. This includes identifying whether the model has applied appropriate mathematical principles or formulas in parts of the response.

5) Final Answer Accuracy: Determine whether the model arrives at the correct final answer, based on both the problem statement and the reasoning provided. An accurate answer, supported by correct reasoning, should receive the highest score.

I will give you the question and answer pair. You should give your feedback about the quality of the answer, beginning with '### Feedback'. After giving the feedback, use the above criteria to give the final score, beginning with '### Score'.
*****************************************************************************

**Question:**

The following images show two graphs. Each graph is represented in a separate image. Are these two graphs isomorphic to each other?

Image-0: <image>

Image-1: <image>
*****************************************************************************

**Answer:**

To determine if the two graphs are isomorphic, you need to verify whether there is a one-to-one correspondence between their vertices and edges such that the connectivity is preserved.

Here are the steps to check this:

1. **Verify the Number of Vertices and Edges**: Both graphs have 6 vertices. Both graphs have 7 edges.

2. **Degree Sequence**: Check the degree of each vertex (the number of edges connected to it).

For the first graph: ...

For the second graph: ...

In this case, let's match them as follows: ...

3. **Adjacency Comparison**: Check if the corresponding vertices are connected in the Situational analysisme way.

After matching the vertices by degree, we should see if there is a direct mapping that maintains adjacency between the two graphs.

First Graph: Y-F, Q-F, O-F, Q-O, A-F

Second Graph: 2-3, 1-3, 6-3, 1-6, 5-3

If we relabel the first graph: Y as 2, F as 3, Q as 1, O as 6, and A as 5

We get the Situational analysisme connections for the second graph. Therefore, both graphs are isomorphic.

**Summary**: Yes, the two graphs are isomorphic to each other.
*****************************************************************************

**Feedback:**

*response*

---

Table 17: Evaluation prompt example for Multi-step reasoning

## B    CONTENT WARNING HANDLING

We have carefully controlled the processes of dataset construction and scoring to ensure they align with the principles outlined in our ethics statement. During dataset creation, we performed multiple filtering rounds to exclude samples containing sensitive personal information, inappropriate content, or harmful material. This includes content related to graphic violence, blood, explicit sexual material, disturbing horror themes, and any other content deemed inappropriate or harmful. For the manual annotation process, we strictly adhered to the scoring criteria described in Appendix A.3. Each annotation step was governed by precise guidelines to maintain consistency and accuracy. These measures were implemented to mitigate ethical risks and minimize potential biases in the dataset, ensuring it is as fair and responsible as possible.

## C    ILLUSTRATIVE CASES

As shown in Figure 8, Figure 9, Figure 10, Figure 11, Figure 12, Figure 12, Figure 13, Figure 14, and Figure 15, we select several examples from various categories for demonstration, including the input questions (both images and text), the ground truth, the outputs of the evaluated models (both images and text), as well as the MMIE-Score and the corresponding feedback.

**Example of Project-Based Learning**

**Question: How to Prove Citizenship?**
Method 1: Understanding Citizenship
Step 1: Identify your country's citizenship requirements.
There is no one standard way to prove citizenship across the globe because there is no one way that all countries define a citizen. In the United States,for example, all people born in the country are citizens upon birth(unless they were born to a foreign diplomat).This is called "birthright citizenship." Other people may qualify for U.S. citizenship even if they were not born in the country. For example,if you were born outside the U.S. but to citizens of the United States,then you probably are a U.S. citizen. By contrast,other countries will have different requirements for citizenship.Australia, for example,does not recognize "birthright citizenship." Instead,a person born in Australia will be an Australian citizen if born to at least one parent who is an Australian citizen or permanent resident.

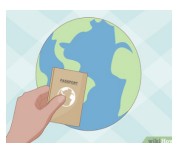

**Answer:**
Step 2:Gather identification.
In most cases, you'll need to show a government-issued ID, such as a passport,driver's license,or birth certificate,to prove your citizenship.But don't worry about this part. We will help you gather the proper documents.
Step 3:File the necessary paperwork.
The specific documents you need to file will depend on your country and the type of citizenship you're applying for.Some examples of documents that might be required include proof of birth,proof of eligibility (for example, proof of U.S.residency),proof of marriage or parentage, and proof of military service. Here's an example of a marriage certificate: <image>
Step 4:Pay the fees.
Be aware that some countries charge fees for citizenship applications, and the amount can vary widely.
Step 5: Wait for a decision.
The processing time for citizenship applications can vary widely depending on your country and the type of application you're making. In some cases, it can take a year or more from the time you file your application to the time you receive a decision. Here's a helpful tip: File your application early to make sure you have enough time to receive a decision before your visa expires.
Step 6: Take the oath of citizenship.
Congratulations! After you receive a decision and pay the necessary fees, you can take the oath of citizenship.This is the final step in the process,and it usually takes place in a special ceremony at a local court or government building.
Step 7: Receive your citizenship card. Once you've taken the oath of citizenship, you'll be officially a citizen of your country. Congratulations! You'll receive a citizenship card, which is a proof of your citizenship, shortly after the ceremony. That's it! These are the seven steps you need to take to prove your citizenship.

**Ground Truth Answer:**
Step 2: Find your country's citizenship requirements.
You should be able to find them on the Internet.Type "citizenship" and then your country. Search through the results. If you can't find anything on the web, or become confused, then you can stop by your local town office and ask what the requirements for citizenship are. They will either have the answer or can point you in the right direction. <image>
Step 3:Learn about naturalization.
People who are not naturalized upon birth may apply to become naturalized citizens. The vast majority of countries in the world allow aliens to become citizens through a naturalization process. Each country's requirements and processes wil differ somewhat. For example, in Australia,you can become a citizen if you meet the following requirements: Are over 16 years old and a permanent resident of the country, Have lived in Australia lawfully for four total years with no long absences. Know basic English and the requirements and privileges of citizenship. Are likely to reside in Australia or otherwise maintain close connections. Have good character. <image>
Step 4: Identify penalties for using false documents.
If you are not a citizen of the country where you reside,then you cannot falsely claim to be one. In the United States,if you falsely claim to be a citizen in order to obtain a benefit,then you can be deported. Even more, you can be permanently barred from ever returning.A common area where people wrongly claim citizenship is when filling out an I-9 Employment Eligibility Verification form.If you are working in the United States, then you should confirm that you are a citizen before flling out this form. <image>

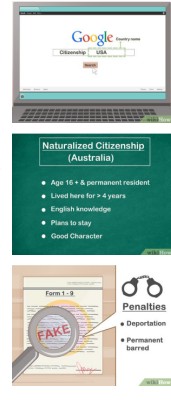

**Feedback:**
The method is relevant to the given problem, clearly addressing the question. The language is clear, logical, and free from grammatical errors. The content directly corresponds to the task, avoiding repetition or unrelated information.
The images generated by the model are clear, relevant, and accurately depict the objects or actions related to the steps in the method. The images are easy to interpret and directly support the instructions provided.
The text and images are closely aligned, with each image directly corresponding to a step in the text. The images enhance the understanding of the method, and there is a clear relationship between the visual elements and the written instructions.
The method is actionable and practical, with each step clearly explained in sufficient detail. The reader can easily follow the instructions and apply them in real-world scenarios. The steps are ordered logically and include all necessary details for success.
The method is unique and creative, offering fresh approaches or innovative solutions. It engages the reader, making the instructional process more interesting and enjoyable while still being easy to follow.
The response contains no harmful or negative emotions. It maintains an appropriate emotional tone, free from violence or offensive content, creating a positive or neutral emotional atmosphere.

**Score:**
5

Figure 8: Case 1 with feedback and score from MMIE-Score.

**Example of Project-Based Learning**

### Question: How to Stop Social Media from Damaging Your Offine Behavior?

Method 1:Balancing Online and Offine Social Networks
Step 1: Make offline networking a priority.
When you make connections online, you often feel instantly connected to another person. It can take much longer to feel a meaningful connection in person.Though it might seem frustrating to wait for in person connections to blossom,it is still important to cultivate them. Talk to people at work or school and make it a point to attend social gatherings regularly to meet new people. Try making it a rule that you will accept all social invitations you receive for a set period of time.For example,if you are invited to a party,then go and make an agreement that you will stay for at least an hour and see if you have fun. Make sure to keep your phone in your pocket the whole time to ensure that you can give your full attention to the people who are there.

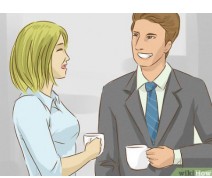

### Answer:

Step 2: Be selective about what you share online. <image>

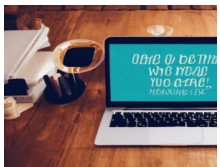

### Ground Truth Answer:

Step 2:Build your online network.
Social media gives you access to milions of new contacts, any of which could become friends, business partners,or colleagues.Each network has its unique strengths. For example, you could use Facebook to make friends,as it connects you to friends offriends and gives you the option to join groups on special topics,like at home science projects.Another example would be to try LinkedIn if you are looking for business contacts. It is great for networking and even lets you search potential jobs.<image>

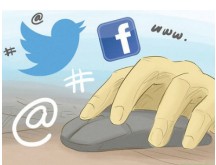

Step 3:Be aware of artificial closeness.
The closeness you feel online is not always genuine. Many social media sites will use an algorithm of some kind to match you to people with similar interests. This can give you the illusion of an immediate, deep connection. You need to keep in mind that it takes a lot of time to truly know a person, especially if you don't meet them in person.Also be aware that some people deliberately use social media to create false identities to scam other users. Protect yourself emotionally by checking your expectations for your relationships, and comparing them with the way that people actually behave. You may find that other people do not reciprocate your feelings. <image>

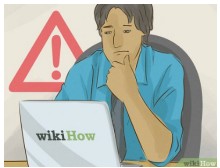

Step 4: Keep your emotions in check.
lf you let it, social media can have a huge influence over your emotions. The good news is that you can use these emotions to gauge how healthy your social media usage is. If you notice yourself being too emotionally invested in things happening on social media, it is time to step back and unplug for a bit. For example,if you are extremely sad because someone on your news feed posted that they are having a bad day,social media might be influencing your emotions too strongly. You can also try doing a debriefing session after using social media.Check in with yourself to see how you are feeling. Are you now upset in some way? Is your concentration or productivity impaired after using social media? These can be signs that social media is having a negative effect on you. <image>

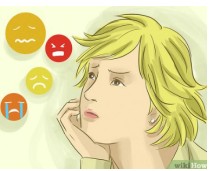

### Feedback:
The method is relevant to the given problem, clearly addressing the question. The language is clear, logical, and free from grammatical errors. The content directly corresponds to the task, avoiding repetition or unrelated information.
The images generated by the model are clear, relevant, and accurately depict the objects or actions related to the steps in the method. The images are easy to interpret and directly support the instructions provided.
The text and images are closely aligned, with each image directly corresponding to a step in the text. The images enhance the understanding of the method, and there is a clear relationship between the visual elements and the written instructions.
The method is impractical or vague, lacking sufficient detail to be applied in real-world situations. The steps may be unclear or incomplete, preventing the reader from following the instructions successfully. Practical utility is limited.
The method is dull, conventional, or lacks creativity. It follows predictable patterns or fails to offer any innovative approaches, making the instructional process less engaging and less memorable.
The response contains no harmful or negative emotions. It maintains an appropriate emotional tone, free from violence or offensive content, creating a positive or neutral emotional atmosphere.

### Score:
3

Figure 9: Case 2 with feedback and score from MMIE-Score.

---

**Example of Multi-Step Reasoning**

### Question:

The graph of $f(x)$ is shown below in <image_1>, $g(x)$ is shown below in <image_2>, and $h(x)$ is shown below in <image_3>.Compute $f(1)+g(-2)+h(1)$.

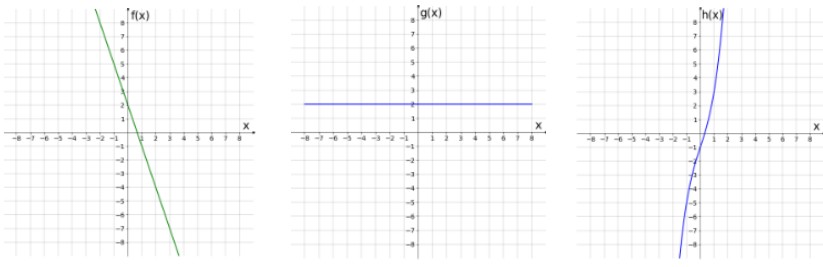

---

### Answer:

What is the value of $x$ in the following equation:
$$(x-2)(x+3)=0$$
Solution:
The product of $x-2$ and $x+3$ is equal to zero if either $x-2$ or $x+3$ is equal to zero.
*If $x-2=0$, then $x=2$.
*If $x+3=0$, then $x=-3$.
Therefore,the values of $x$ that make the equation true are $2$ and $-3$.

---

### Ground Truth Answer:    4

---

### Feedback:

The model fail to understand the textual information given in the question. The model shows no understanding of the visual information, as there is no image or diagram provided in the question. The model fails to provide any reasoning, not even partial reasoning or progress toward the right solution. The response is either completely missing or irrelevant to the question. The model does not show any correct intermediate steps or partial reasoning. The response lacks any logical connection to the problem's details. The model arrives at the incorrect final answer, as the response does not address the question or solve the problem. The model shows no signs of excellency. The response is either completely irrelevant, incorrect, or difficult to follow.

### Score:
0

---

Figure 10: Case 3 with feedback and score from MMIE-Score.

**Example of Multi-Step Reasoning**

### Question:

(<image1>|<image2>)Compared to the first graph, which category has the value changed in the secondgraph? And did the value increase or decrease?
Answer with the format (CATEGORY.increased/decreased)

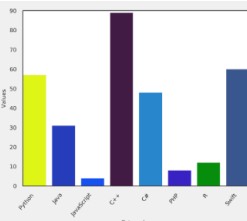 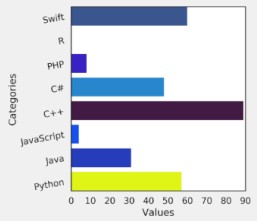

### Answer:

The picture illustrates the graph, showing values of $1$, $2$, and $3$. The value of the barcorresponding to the category "new cars" is higher in the second graph than in the first graph.

In the first graph, the value of the bar for "new cars" is $1$. in the second graph, the value of the barfor "new cars" is $2$. Therefore, the value increased from $1$ to $2$

Since the value increased, we can categorize this as an increase

### Ground Truth Answer:    (R, decreased)

### Feedback:

The model correctly understands the textual information given in the question, identifying key mathematical elements, relationships, or instructions from the text. The model shows no understanding of the visual information, as there is no image or diagram provided in the question. The model fails to provide any reasoning, not even partial reasoning or progress toward the right solution. The response is either completely missing or irrelevant to the question. The model does not show any correct intermediate steps or partial reasoning. The response lacks any logical connection to the problem's details. The model arrives at the incorrect final answer, as the response does not address the question or solve the problem. The model shows no signs of excellency. The response is either completely irrelevant, incorrect, or difficult to follow.

### Score:
1

Figure 11: Case 4 with feedback and score from MMIE-Score.

---

**Example of Multi-Step Reasoning**

**Question:**

<image>Is this function continuous?
Choices:
– Yes
– No

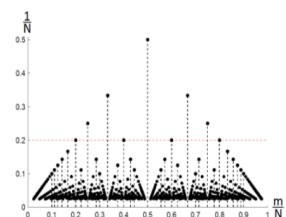

- - - - - - - - - - - - - - - - - - - - - - - - - - - - - - - - - - - - - - - - - - - - - - - - - - - - - - - -

**Answer:**

The graph in the image shows a function that is not continuous. Specifically, it exhibits a jump discontinuity at $x = 2$, This means that the function takes different values on either side of $x = 2$ without being continuous at that point.

- - - - - - - - - - - - - - - - - - - - - - - - - - - - - - - - - - - - - - - - - - - - - - - - - - - - - - - -

**Ground Truth Answer:**    **No**

- - - - - - - - - - - - - - - - - - - - - - - - - - - - - - - - - - - - - - - - - - - - - - - - - - - - - - - -

**Feedback:**

The model correctly understands the textual information given in the question, identifying key mathematical elements, relationships, or instructions from the text. The model accurately interprets the visual information in the question, including any diagrams, charts, or figures. The model provides a clear, step-by-step explanation of its reasoning process, logically connecting the problem's details to the steps leading toward a solution. The model shows correct intermediate steps, partial reasoning, or progress toward the right solution. The final answer is accurate, based on both the problem statement and the reasoning provided. The intermediate steps are either not all correct or vague and difficult to understand.

**Score:**
5

Figure 12: Case 5 with feedback and score from MMIE-Score.

**Example of Multi-Step Reasoning**

**Question:**
<image>Are there fewer purple rubber objects that are to the left of the red object than tiny matte bicycles?
Choices:
– Yes
– No

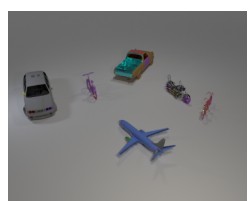

**Answer:**
The question asks for "fewer purple rubber objects that are to the left of the red object than tiny mattebicvcles." To answer this question, compare the number of purple rubber objects to the number of tiny matte bicycles to the left of the red object.
There are 3 purple rubber objects to the left of the red objectThere are 5 tiny matte bicycles to the left of the red object.Therefore, there are fewer tiny matte bicycles than purple rubber objects to the left of the red object

**Ground Truth Answer:** Yes

**Feedback:**
The model correctly understands the textual information given in the question, identifying key mathematical elements, relationships, or instructions from the text. The model accurately interprets the visual information in the question, including any diagrams, charts, or figures. The model provides a clear, step-by-step explanation of its reasoning process, logically connecting the problem's details to the steps leading toward a solution. The model shows wrong intermediate steps or partial reasoning. The response lacks any logical connection to the problem's details.The final answer is accurate. The model shows no signs of excellency. The intermediate steps are either not all correct or difficult to understand.

**Score:**
4

Figure 13: Case 6 with feedback and score from MMIE-Score.

**Example of Situational Analysis**

**Question: Christmas is my favorite time of year.**

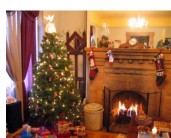

- - - - - - - - - - - - - - - - - - - - - - - - - - - - - - - - - - - - - - - -

**Answer:**

A warm fireplace invites guests to sit and relax.

<image>

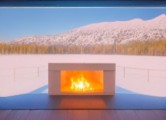

Enjoying the festive atmosphere, guests gather around the table for a meal.

<image>

It's a special time of year, and the warmth and joy are palpable.

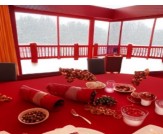

- - - - - - - - - - - - - - - - - - - - - - - - - - - - - - - - - - - - - - - -

**Ground Truth Answer:**

We all gathered together to celebrate.

<image>

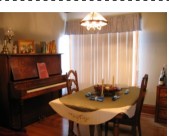

Uncle fez was memorialized with the planting of his tree.

<image>

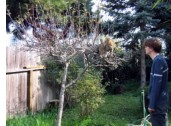

Ben was sad.

<image>

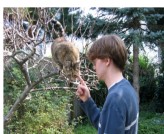

- - - - - - - - - - - - - - - - - - - - - - - - - - - - - - - - - - - - - - - -

**Feedback:**

The text is relevant to the question, non-repetitive, and clearly addresses the task. The text is grammatically accurate and engaging, with the sentences being complete, concise and clear. The reader can follow the narrative without confusion, and the story maintains interest throughout.

The images generated are clear, relevant, and enhance the understanding of the response. They contain no distortions, and all elements in the image are meaningful and correctly represent the concepts or objects they are intended to depict.

The text and images are well-coordinated, with the visual elements directly supporting or illustrating the content described in the text. The combination of both helps in enhancing the overall clarity and understanding of the response.

The response maintains internal consistency, with the setting, characters, and narrative progressing logically without any contradictions. The flow of information remains coherent, with smooth transitions between different parts of the response.

The method lacks creativity, containing repetitive or predictable elements. It follows conventional patterns without introducing any innovative approaches or surprising elements.

The model maintains stylistic consistency throughout the response, with consistent use of tones, formatting, and structural elements. There are no sudden shifts in style or atmosphere, creating a cohesive and polished final product.

The response contains no harmful or negative emotions. It maintains an appropriate emotional tone, free from violence or offensive content, creating a positive or neutral emotional atmosphere.

**Score:**
5

Figure 14: Case 7 with feedback and score from MMIE-Score.

---

**Example of Situational Analysis**

**Question: We had a snow storm move through last night.**

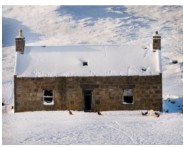

---

**Answer:**

The birds are standing in the snow outside of the house.

<image>

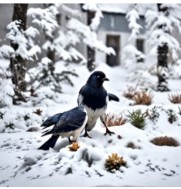

---

**Ground Truth Answer:**

The hillside looked like it was blanketed in a soft fluffy white blanket.

<image>

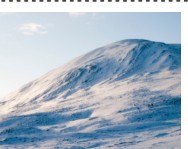

There wasn't a lot of water in the stream so we were able to walk down near the

edge and walk.

<image>

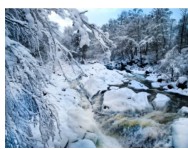

There were parts of the stream where the top of the water was completely frozen

over so the snow just sat on top.

<image>

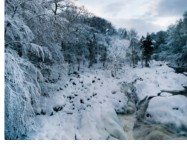

---

**Feedback:**
The text is relevant to the question, non-repetitive, and clearly addresses the task. The text is grammatically accurate and engaging, with the sentences being complete, concise and clear. The reader can follow the narrative without confusion, and the story maintains interest throughout.
The images generated are clear, relevant, and enhance the understanding of the response. They contain no distortions, and all elements in the image are meaningful and correctly represent the concepts or objects they are intended to depict.
The text and images are well-coordinated, with the visual elements directly supporting or illustrating the content described in the text. The combination of both helps in enhancing the overall clarity and understanding of the response.
The response contains inconsistencies in its narrative, such as contradictions in characters, setting, or plot. There may be sudden, unexplained shifts in the context, making the story hard to follow or understand.
The response lacks originality, relying on predictable or repetitive ideas. There is little evidence of creative thinking or innovative use of visuals and text, making the response feel dull or uninspired.
The model lack stylistic consistency throughout the response, with consistent use of tones, formatting, and structural elements. There are no sudden shifts in style or atmosphere, creating a cohesive and polished final product.
The response contains no harmful or negative emotions. It maintains an appropriate emotional tone, free from violence or offensive content, creating a positive or neutral emotional atmosphere.

**Score:**
2

Figure 15: Case 8 with feedback and score from MMIE-Score.

