# OpenReview forum: "MMIE: Massive Multimodal Interleaved Comprehension Benchmark for Large Vision-Language Models"
_ICLR.cc/2025/Conference — ICLR 2025 Oral_

### Official Review · Reviewer_t4WL · 2024-11-01

**Soundness:** 3
**Presentation:** 3
**Contribution:** 2
**Rating:** 8
**Confidence:** 4

**Summary:**

This paper introduces MMIE, a large-scale benchmark designed to evaluate interleaved multimodal comprehension and generation capabilities of Large Vision-Language Models (LVLMs). MMIE comprises 20K multimodal queries across 12 fields, supporting interleaved text and multi-image inputs and outputs in both multiple-choice and open-ended formats. It provides a comprehensive framework to assess LVLMs on complex, real-world tasks that demand high multimodal reasoning and synthesis skills.

**Strengths:**

1. MMIE provides a robust evaluation framework for Large Vision-Language Models (LVLMs) by supporting cross-modal, interleaved input and output of both text and multiple images. This flexibility in handling multi-image inputs and outputs, along with the inclusion of multiple-choice and open-ended question formats, broadens the range and depth of tasks LVLMs are tested on. Furthermore, the large data scale and comprehensive scope of MMIE allow it to evaluate LVLMs in a way that captures the complexities of real-world multimodal interactions, offering a thorough assessment of these models’ interleaved multimodal comprehension and generation capabilities.

2. MMIE introduces an automated scoring model fine-tuned with human-annotated data, developed using detailed evaluation criteria. This approach addresses the challenge of bias often seen in traditional metrics, enhancing the objectivity and precision of multimodal model evaluation.

**Weaknesses:**

1. Lack of fine-grained presentation of results across fields: Although the paper evaluates several models, it does not fully present results across different fields, making it challenging to analyze performance gaps in specific domains. This limitation may restrict insights into targeted improvements for specific models in particular fields.

2. Potential Overfitting to Specific Benchmark Tasks: Since the scoring model in MMIE is fine-tuned on data specific to the benchmark, it may overfit to the types of tasks and response styles within MMIE.   This could limit its generalizability to other multimodal datasets or tasks, potentially reducing its effectiveness when evaluated in new contexts outside of MMIE’s scope.

**Questions:**

Given that the scoring model is trained on data similar to MMIE (12 domains), I am curious about its generalizability to other benchmarks. Does the scoring model work on other benchmark data, such as MMLU, which includes 30 domains? This would provide insight into whether the model’s evaluation criteria are adaptable to a broader range of multimodal tasks.


Edit: I have read the author response and revised my review.

---

> ### Author Response · Authors · 2024-11-21
> **Response to Reviewer t4WL (1/3)**
>
> Thank you for your constructive comments and suggestions. We have revised our paper according to your comments. We respond to your questions below and would appreciate it if you could let us know if our response addresses your concerns.
>
> ****
>
> >**Q1**: Lack of fine-grained presentation of results across fields: Although the paper evaluates several models, it does not fully present results across different fields, making it challenging to analyze performance gaps in specific domains. This limitation may restrict insights into targeted improvements for specific models in particular fields.
>
> **A1**: Following your advice, as shown in Table R1, we provide detailed results for the model's performance across the following 12 fields. In the Project-based learning (PBL) category, among the seven fields, most models exhibit slightly lower performance in finance and education, while achieving better results in art, sports, and philosophy. In the Multi-step reasoning (MSR) category, across the four fields, models generally perform worse in coding and statistics but demonstrate stronger performance in mathematics and physics. We have revised the paper and put these details in Appendix A.7.

---

> ### Author Response · Authors · 2024-11-21
> **Response to Reviewer t4WL (2/3)**
>
> **Table R1**: Performance comparison across various fields for different LVLMs. Since the Multi-step reasoning category (i.e., mathematics, physics, coding, statistics) do not require image outputs, the scores of integrated LVLMs for this category remain consistent.
>
> | **Model**             | **Art** | **EECS** | **Education** | **Finance** | **Health** | **Philosophy** | **Sports** | **Literature** | **Mathematics** | **Physics** | **Coding** | **Statistics** |
> |------------------------|---------|----------|---------------|-------------|------------|----------------|------------|----------------|-----------------|-------------|------------|----------------|
> | MiniGPT-5             | 58.90   | 58.13    | 53.17         | 53.47       | 53.00      | 54.18          | 59.80      | 47.63          | 46.53           | 42.75       | 40.24      | 39.17          |
> | Emu-2                 | 50.23   | 48.65    | 43.90         | 44.55       | 45.52      | 42.47          | 48.35      | 39.65          | 54.93           | 51.10       | 47.66      | 49.31          |
> | Gill                  | 61.27   | 60.60    | 53.12         | 55.80       | 58.92      | 64.53          | 63.67      | 46.72          | 38.11           | 42.73       | 38.05      | 38.43          |
> | Anole                 | 58.37   | 61.95    | 59.93         | 56.62       | 58.73      | 65.95          | 59.10      | 48.95          | 47.37           | 53.92       | 49.81      | 55.79          |
> | GPT+OpenJourney       | 74.30   | 74.12    | 70.92         | 69.77       | 70.07      | 79.17          | 71.98      | 53.05          | 58.07           | 51.63       | 52.60      | 52.37          |
> | GPT+SD3               | 74.02   | 74.97    | 70.72         | 66.23       | 72.88      | 75.00          | 72.93      | 53.00          | 58.07           | 51.63       | 52.60      | 52.37          |
> | GPT+SDXL              | 74.87   | 72.57    | 75.05         | 71.87       | 73.75      | 66.67          | 70.65      | 56.12          | 58.07           | 51.63       | 52.60      | 52.37          |
> | GPT+Flux              | 74.37   | 69.83    | 72.10         | 63.15       | 67.63      | 79.17          | 69.88      | 54.97          | 58.07           | 51.63       | 52.60      | 52.37          |
> | Gemini+OpenJourney    | 68.47   | 65.28    | 67.48         | 67.35       | 68.67      | 76.18          | 73.65      | 48.08          | 61.83           | 65.02       | 55.85      | 57.49          |
> | Gemini+SD3            | 69.78   | 67.17    | 69.57         | 64.02       | 71.87      | 83.33          | 72.55      | 47.48          | 61.83           | 65.02       | 55.85      | 57.49          |
> | Gemini+SDXL           | 73.25   | 66.07    | 73.65         | 70.07       | 74.70      | 76.18          | 75.20      | 49.43          | 61.83           | 65.02       | 55.85      | 57.49          |
> | Gemini+Flux           | 72.22   | 62.32    | 70.23         | 62.42       | 72.93      | 83.33          | 74.72      | 47.07          | 61.83           | 65.02       | 55.85      | 57.49          |
> | LLaVA+OpenJourney     | 75.80   | 72.95    | 73.45         | 71.53       | 74.50      | 83.33          | 74.03      | 54.12          | 47.81           | 49.63       | 46.21      | 45.47          |
> | LLaVA+SD3             | 75.40   | 73.92    | 70.88         | 69.03       | 75.37      | 83.33          | 74.02      | 54.72          | 47.81           | 49.63       | 46.21      | 45.47          |
> | LLaVA+SDXL            | 76.85   | 72.10    | 74.42         | 74.70       | 75.08      | 83.33          | 75.18      | 55.97          | 47.81           | 49.63       | 46.21      | 45.47          |
> | LLaVA+Flux            | 76.00   | 69.90    | 72.60         | 64.63       | 75.38      | 83.33          | 72.28      | 54.23          | 47.81           | 49.63       | 46.21      | 45.47          |
> | Qwen+OpenJourney      | 75.27   | 68.92    | 70.57         | 70.62       | 72.13      | 80.95          | 75.85      | 52.73          | 57.29           | 59.52       | 54.05      | 51.66          |
> | Qwen+SD3              | 76.72   | 72.12    | 70.85         | 68.47       | 72.87      | 76.18          | 74.82      | 54.98          | 57.29           | 59.52       | 54.05      | 51.66          |
> | Qwen+SDXL             | 75.12   | 72.57    | 73.25         | 72.68       | 73.30      | 69.05          | 77.78      | 52.58          | 57.29           | 59.52       | 54.05      | 51.66          |
> | Qwen+Flux             | 77.07   | 68.33    | 70.32         | 61.93       | 70.67      | 73.80          | 76.67      | 54.23          | 57.29           | 59.52       | 54.05      | 51.66          |

---

> ### Author Response · Authors · 2024-11-21
> **Response to Reviewer t4WL (3/3)**
>
> >**Q2**: Potential Overfitting to Specific Benchmark Tasks: Since the scoring model in MMIE is fine-tuned on data specific to the benchmark, it may overfit to the types of tasks and response styles within MMIE. This could limit its generalizability to other multimodal datasets or tasks, potentially reducing its effectiveness when evaluated in new contexts outside of MMIE’s scope. Given that the scoring model is trained on data similar to MMIE (12 domains), I am curious about its generalizability to other benchmarks. Does the scoring model work on other benchmark data, such as MMLU, which includes 30 domains? This would provide insight into whether the model’s evaluation criteria are adaptable to a broader range of multimodal tasks.
>
> **A2**: ​​Thank you very much for your suggestions. Our scoring model is specifically designed for the interleaved benchmark and fine-tuned on the data we collected for our specific task. ​​Following your suggestion, we apply MMIE-Score to an OOD dataset to validate its generalization capability. However, the recommended MMLU is a QA dataset in text-only format, which is not applicable to our evaluation model. Similarly, the recommended MMMU is a VQA dataset with multiple-choice questions, where accuracy can be directly calculated, making it also unsuitable for our evaluation model. Instead, we conduct an experiment using 100 examples from the COCO dataset [1], a widely used image captioning dataset for natural image scenarios. We use MMIE-Score to evaluate the generated text descriptions based on input images, using Text Quality, Image-Caption Relevance, Contextual Consistency, Diversity, Specificity and Detail, and Stylistic Consistency and Correspondence as a 6-point scoring criteria, with the Emotional Impact for penalty. We also compare the scoring quality of GPT-4o, InternVL-2 and Anole. The correlation results between all scoring methods and human evaluations are shown in Table R2. MMIE-Score demonstrates excellent performance, significantly outperforming most models and ranking only behind GPT-4o, indicating good generalization capability.
>
> In the future version, we aim to further optimize the training of the MMIE-Score model by expanding the dataset size, incorporating techniques to improve out-of-distribution robustness, and leveraging GPT-4o for data augmentation to enhance the model's cross-task generalization capabilities. We have revised the paper and put these details in Appendix A.8.
>
> **Table R2**: Comparison of scoring LVLMs in COCO benchmark.
>
> | **Models**            | **Cosine Similarity ↑** | **MSE ↓** | **MAE ↓** | **Pearson ↑** |
> |------------------------|-------------------------|-----------|-----------|---------------|
> | InternVL-2.0-4B       | 0.489                  | 16.042    | 3.255     | 0.062         |
> | Anole                 | 0.544                  | 8.663     | 3.190     | 0.058         |
> | GPT-4o                | **0.745**              | **3.596** | **1.734** | 0.097       |
> | MMIE-Score (Ours) | 0.673                | 5.116   | 2.380   | **0.104**     |
>
>
> [1] Lin T Y, Maire M, Belongie S, et al. Microsoft coco: Common objects in context. ECCV, 2014

---

> > ### Comment · Reviewer_t4WL · 2024-11-26
> >
> > Thank you for your detailed reply.
> >
> > Thank you for adding results across various fields for different LVLMs, which I believe will provide some new insights for subsequent studies.
> >
> > Overall, I think this paper is useful and would revise my review.

---

> > > ### Author Response · Authors · 2024-11-26
> > >
> > > Dear Reviewer t4WL,
> > >
> > > We sincerely thank you for your detailed and insightful feedback, we are happy to see that our response address your concerns. Thanks again for your valuable comments to help us improve our paper.

---

### Official Review · Reviewer_pFfk · 2024-11-03

**Soundness:** 4
**Presentation:** 4
**Contribution:** 3
**Rating:** 8
**Confidence:** 4

**Summary:**

The paper introduces MMIE (Massive Multimodal Interleaved Evaluation), a large-scale benchmark designed to evaluate interleaved multimodal comprehension and generation in Large Vision-Language Models (LVLMs). MMIE comprises 20K multimodal queries across 12 fields, including math, coding, physics, literature, health, and arts. The benchmark supports both interleaved text and image inputs and outputs, including image generation, offering a mix of multiple-choice and open-ended question formats. As evaluation metric, the authors propose an finetuned (multimodal) LLM as a scoring model. This metric aims to reduce bias and improve evaluation accuracy over existing LLMs as-a-judges. The results reveal that current LLMs have significant room for improvement when evaluated on MMIE.

**Strengths:**

* Originality: The paper introduces a novel and comprehensive benchmark for evaluating interleaved multimodal comprehension and generation, addressing an underexplored but increasingly important area.
* Quality: The benchmark is large-scale and diverse, covering a wide range of domains and task formats, which enhances its utility and applicability. The methodology for dataset curation and quality control is rigorous.
* Automated Evaluation Metric: The proposed scoring model provides a more reliable and unbiased evaluation compared to using a "raw" GPT model, strengthening the analysis beyond traditional metrics.
* Human Scoring Comparison: Comparison of different scoring model against human annotations is great in highlighting the strength.

**Weaknesses:**

* Potential Biases in Scoring Model: The reliance on a LLM as a scoring model may introduce biases inherent in the base model. A more thorough analysis of these potential biases, and how they might affect evaluation outcomes across different domains or tasks, might be needed.
* Content Warning Handling: The paper includes a content warning but does not elaborate on it, as Ethical Statement highlight strict guidelines and lack of bias in the dataset.

**Questions:**

* Dataset Size: The 20K sample benchmark is substantial and statistically robust, enabling a comprehensive evaluation. However, validating the full dataset with 20K samples may be costly. Have the authors considered designating a "mini" subset for consistent yet quicker evaluation? For example, using 800 samples selected for scoring model training.
* Visual Component Importance: Is it feasible to analyze the impact of visual content within the benchmark? For instance, comparing text-only generation with generation that includes images. Some examples in Figure 1 do not rely on image content but rather use images solely for illustrative purposes.

---

> ### Author Response · Authors · 2024-11-21
> **Response to Reviewer pFfk (1/4)**
>
> Thank you for reviewing our paper and for your valuable feedback. Below, we address your concerns point by point and we’ve revised our paper according to your suggestions. We would appreciate it if you could let us know whether your concerns are addressed by our response.
>
> ****
>
> >**Q1**: Potential Biases in Scoring Model: The reliance on a LLM as a scoring model may introduce biases inherent in the base model. A more thorough analysis of these potential biases, and how they might affect evaluation outcomes across different domains or tasks, might be needed.
>
> **A1**: We present several comparison results of our scoring model with other baselines across different categories and model types (Interleaved and Integrated LVLMs). As shown in Table R1, the results show that although our model exhibits slightly varying performance across different categories, for example Cosine Similarity scores are higher for Situational Analysis (SA) and Multi-Step Reasoning (MSR) category and Pearson scores are higher for Project-Based Learning (PBL) category, the biases remain little. Overall, our MMIE-Score consistently outperforms other baselines. We have revised the paper and put these details in Appendix A.4.

---

> ### Author Response · Authors · 2024-11-21
> **Response to Reviewer pFfk (2/4)**
>
> **Table R1**: Comparison of scoring LVLMs and traditional image-text alignment metrics across different models and categories.
>
> | **Category**                      | **Models**               | **Cosine Similarity ↑** | **MSE ↓**   | **MAE ↓**   | **Pearson ↑** |
> |-----------------------------------|--------------------------|--------------------------|-------------|-------------|---------------|
> | **Situational Analysis (Interleaved)** | Text-Image CLIPScore     | 0.604                    | 6.710       | 2.057       | 0.022         |
> |                                   | InternVL-2.0-4B          | 0.691                    | 14.001      | 3.382       | 0.094         |
> |                                   | Anole                    | 0.867                    | 3.973       | 1.579       | 0.045         |
> |                                   | GPT-4o                   | 0.718                    | 4.195       | 1.573       | 0.042         |
> |                                   | **MMIE-Score (Ours)**    | **0.895**                | **3.547**   | **1.502**   | **0.098**     |
> | **Project-based Learning (Interleaved)** | Text-Image CLIPScore     | 0.612                    | 7.669       | 2.197       | 0.022         |
> |                                   | InternVL-2.0-4B          | 0.654                    | 16.560      | 3.499       | 0.072         |
> |                                   | Anole                    | 0.689                    | 4.423       | 1.566       | 0.047         |
> |                                   | GPT-4o                   | 0.661                    | 3.837       | 1.670       | 0.045         |
> |                                   | **MMIE-Score (Ours)**    | **0.760**                | **3.163**   | **1.496**   | **0.114**     |
> | **Multi-step Reasoning (Interleaved)** | Text-Image CLIPScore     | -                        | -           | -           | -             |
> |                                   | InternVL-2.0-4B          | 0.770                    | 16.050      | 3.393       | 0.084         |
> |                                   | Anole                    | 0.739                    | 3.612       | 1.615       | 0.054         |
> |                                   | GPT-4o                   | 0.798                    | 3.985       | 1.674       | 0.045         |
> |                                   | **MMIE-Score (Ours)**    | **0.814**                | **3.767**   | **1.347**| **0.106**     |
> | **Situational Analysis (Integrated)** | Text-Image CLIPScore     | 0.640                    | 7.701       | 2.184       | 0.023         |
> |                                   | InternVL-2.0-4B          | 0.695                    | 13.960      | 3.432       | 0.073         |
> |                                   | Anole                    | 0.823                    | 4.222       | 1.408       | 0.051         |
> |                                   | GPT-4o                   | 0.763                    | 3.707       | 1.521       | 0.039         |
> |                                   | **MMIE-Score (Ours)**    | **0.843**                | **2.811**   | **1.384**   | **0.093**     |
> | **Project-based Learning (Integrated)** | Text-Image CLIPScore     | 0.551                    | 7.214       | 1.990       | 0.023         |
> |                                   | InternVL-2.0-4B          | 0.759                    | 15.250      | 3.075       | 0.094         |
> |                                   | Anole                    | 0.691                    | 4.295       | 1.718       | 0.050         |
> |                                   | GPT-4o                   | 0.748                    | 3.958       | 1.782       | 0.044         |
> |                                   | **MMIE-Score (Ours)**    | **0.795**                | **3.388**   | **1.432**   | **0.122**     |
> | **Multi-step Reasoning (Integrated)** | Text-Image CLIPScore     | -                        | -           | -           | -             |
> |                                   | InternVL-2.0-4B          | 0.688                    | 17.451      | 3.590       | 0.092         |
> |                                   | Anole                    | 0.713                    | 3.713       | 1.727       | 0.054         |
> |                                   | GPT-4o                   | 0.753                    | 3.594       | 1.491       | 0.036         |
> |                                   | **MMIE-Score (Ours)**    | **0.825**                | **3.112**   | **1.583**   | **0.097**     |

---

> ### Author Response · Authors · 2024-11-21
> **Response to Reviewer pFfk (3/4)**
>
> >**Q2**: Content Warning Handling: The paper includes a content warning but does not elaborate on it, as Ethical Statement highlights strict guidelines and lack of bias in the dataset.
>
> **A2**: We have carefully controlled the processes of dataset construction and scoring to ensure they align with the principles outlined in our ethics statement. During dataset creation, we performed multiple filtering rounds to exclude samples containing sensitive personal information, inappropriate content, or harmful material. This includes content related to graphic violence, blood, explicit sexual material, disturbing horror themes, and any other content deemed inappropriate or harmful. For the manual annotation process, we strictly adhered to the scoring criteria described in Appendix A.3. Each annotation step was governed by precise guidelines to maintain consistency and accuracy. These measures were implemented to mitigate ethical risks and minimize potential biases in the dataset, ensuring it is as fair and responsible as possible. We have revised the paper and put these details in Appendix B.
>
> ****
>
> >**Q3**: Dataset Size: The 20K sample benchmark is substantial and statistically robust, enabling a comprehensive evaluation. However, validating the full dataset with 20K samples may be costly. Have the authors considered designating a "mini" subset for consistent yet quicker evaluation? For example, using 800 samples selected for scoring model training.
>
> **A3**: Thanks for your valuable advice. The fully 20k sample size surely ensures the comprehensiveness of our dataset, but it does come with time costs. Following your advice, we resample evenly from each category and field, and construct a subset of 1000 samples. We re-ran the models on this subset and used MMIE-Score for scoring. As shown in the Table R2 and Table R3, the performance of the models on our subset is consistent with the results from the full dataset. We have revised the paper and put these details in Appendix A.5.
>
> **Table R2**: Performance of the four open-source LVLMs supporting interleaved image-and-text input and output on MMIE’s 1K subset.
>
> | **Model**      | **Situational analysis** | **Project-based learning** | **Multi-step reasoning** | **AVG**    |
> |-----------------|--------------------------|----------------------------|---------------------------|------------|
> | MiniGPT-5       | 46.26                   | 56.53                      | 46.06                    | 52.09      |
> | EMU-2           | 34.44                   | 52.81                      | 48.91                    | 47.54      |
> | GILL            | 48.48                   | 59.49                      | 35.88                    | 52.50      |
> | Anole       | **50.26**               | **60.70**                  | **50.11**                | **56.20**  |

---

> ### Author Response · Authors · 2024-11-21
> **Response to Reviewer pFfk (4/4)**
>
> **Table R3**: Comparison with integrated LVLMs, evaluated on MMIE’s 1K subset. *: LLaVA only supports single-image input and all multi-image queries are thus skipped.
>
> | **LVLM**          | **T2I Model**   | **Situational analysis** | **Project-based learning** | **Multi-step reasoning** | **AVG**    |
> |--------------------|-----------------|--------------------------|----------------------------|---------------------------|------------|
> | **GPT-4o**        | Openjourney     | 56.00                   | 67.81                      | 56.70                    | 62.87      |
> |                   | SD-3            | 51.04                   | 69.48                      | 56.70                         | 62.59      |
> |                   | SD-XL           | 55.07                   | 73.92                      | 56.70                         | 66.13      |
> |                   | Flux            | **57.25**               | 70.69                      | 56.70                         | 64.83      |
> | **Gemini-1.5**    | Openjourney     | 48.25                   | 71.16                      | **60.98**                | 63.63      |
> |                   | SD-3            | 45.88                   | 70.12                      | **60.98**                         | 62.44      |
> |                   | SD-XL           | 47.53                   | 73.40                      | **60.98**                         | 64.73      |
> |                   | Flux            | 47.23                   | 70.05                      | **60.98**                         | 62.74      |
> | **LLaVA-34b**     | Openjourney     | 54.84                   | 72.87                      | 47.58*                   | 63.83      |
> |                   | SD-3            | 56.35                   | 72.71                      | 47.58*                         | 64.12      |
> |                   | SD-XL           | 53.52                   | **77.79**                  | 47.58*                         | 66.31      |
> |                   | Flux            | 55.17                   | 68.27                      | 47.58*                         | 61.29      |
> | **Qwen2-VL-72b**  | Openjourney     | 54.46                   | 72.49                      | 56.69                    | 65.16      |
> |                   | SD-3            | 52.25                   | 74.98                      | 56.69                         | 66.03      |
> |                   | SD-XL           | 54.45                   | 75.03                      | 56.69                         | **66.61**  |
> |                   | Flux            | 55.76                   | 67.19                      | 56.69                         | 62.46      |
>
> ****
>
> >**Q4**: Visual Component Importance: Is it feasible to analyze the impact of visual content within the benchmark? For instance, comparing text-only generation with generation that includes images. Some examples in Figure 1 do not rely on image content but rather use images solely for illustrative purposes.
>
> **A4**:
> Our dataset curation and filtering process ensures that all images included in the examples contribute meaningfully to the overall task. For instance, in tasks like visual storytelling, even when images serve only an illustrative purpose, they still impact the overall output quality. We conduct a comparative experiment to evaluate the difference in performance between interleaved generation (text and images) and text-only generation for the same input(GPT-4o + SDXL) and same evaluated model. The evaluation is scored using MMIE-Score and GPT-4o. As shown in Table R4, results showed that when the model outputs included both text and images, the overall quality was superior to text-only outputs. This ensures that the inclusion of images follows reasonable and well-defined criteria. We have revised the paper and put these details in Appendix A.6.
>
>
> **Table R4**: Comparison of GPT-4o + SDXL’s average score with and without image generation, evaluated by GPT-4o and MMIE-Score.
>
> | **Model**             | **w/o image generation** | **w/ image generation** |
> |------------------------|--------------------------|--------------------------|
> | GPT-4o                | 60.90                   | **71.24**               |
> | MMIE-Score (Ours) | 53.46                   | **65.47**               |

---

### Official Review · Reviewer_eNcJ · 2024-11-04

**Soundness:** 3
**Presentation:** 3
**Contribution:** 3
**Rating:** 8
**Confidence:** 4

**Summary:**

The paper presents MMIE: a benchmark for evaluating interleaved multimodal comprehension and generation abilities of Multimodal LLMs. The evaluation dataset is also publicly released. Further, they propose an automated evaluation metric, using a finetuned LLM. They evaluate various interleaved and “integrated” (text-generation followed to text-to-image generation) LLMs on their proposed benchmark.

**Strengths:**

The MMIE benchmark presented in the paper offers a significant contribution in the interleaved comprehension and generation domain. The benchmark is of significant scale (~20k examples) and well categorised, specifically the project-based learning, situational analysis and multi-step reasoning categories. They also describe in good detail the process of the benchmark creation.

Another significant strength is the automated evaluation metric based on a finetuned LLM. With open-ended evals for interleaved generation, the challenge lies in capturing the various facets including image-text alignment, image quality and text quality. Their proposed method captures these, making it a strong contribution.

The experiment section of the paper is quite detailed. Especially the creation of the integrated LLMs where they combine state-of-the-art LLMs with text-to-image generation models. Sec 5 “Error Analysis” where they identify the typical types of errors offers a great analysis of failure cases.

The MMIE benchmark along with the finetuned model for evaluation is publicly released.

**Weaknesses:**

While the proposed automated evaluation metric using a finetuned LLM is novel and promising, details on the construction of the dataset used for finetuning are missing in the paper.

The claim L110 “The proposed scoring model ... has proven to be comparable to human evaluation.” is not well justified in its current form. For instance, while Table 5 shows that their proposed method has better similarity with human scoring, further details on how the human annotations were obtained are missing.

**Questions:**

1. The effectiveness of the proposed evaluation metric is a function of the evaluation model. Please provide further details on the finetuning method and dataset utilised for the evaluation model.
2. To better understand the role that the evaluation model plays in the pipeline, please provide qualitative examples of eval model responses corresponding to Fig 7,9,10,11,12 in the appendix.
3. Please provide details on how the human annotations were obtained for Table 5 to better support the claim in L110.

Edit: I have read the author response and revised my score.

---

> ### Author Response · Authors · 2024-11-21
> **Response to Reviewer eNcJ**
>
> Thank you for your valuable feedback to help us improve our paper. We have revised our paper based on your feedback. We detail our response below and please kindly let us know if our response addresses your concerns.
>
> >**Q1**: While the proposed automated evaluation metric using a fine-tuned LLM is novel and promising, details on the construction of the dataset used for fine-tuning are missing in the paper.
>
> **A1**: We randomly select examples from each field for human annotation, including the original inputs (images and questions), ground truth, and responses from evaluation models. For each category, we develop comprehensive and detailed criteria with scoring standards. We first annotate 20 examples, providing specific examples for each score as references. To facilitate the annotation process, we design a graphical annotation tool. Finally, we create a dataset of 800 examples with evaluation scores through human annotation, for fine-tuning the scoring model. We have revised the paper and put these details in Appendix A.3.
>
> ****
>
> >**Q2**: The claim L110 “The proposed scoring model ... has proven to be comparable to human evaluation.” is not well justified in its current form. For instance, while Table 5 shows that their proposed method has better similarity with human scoring, further details on how the human annotations were obtained are missing.
>
> **A2**:  We use the same construction method as the fine-tuning dataset to randomly select 200 samples with human-annotated scores for evaluating MMIE-Score. We then implement traditional image-text consistency metrics (e.g., CLIPScore) and LVLM-generated scores, then evaluate their scoring quality using several correlation metrics (e.g., Cosine Similarity, MSE, MAE, Pearson) based on human-annotated scores. As shown in Table 5, MMIE-Score outperforms all the above metrics, maintaining the best performance across the board. We have revised the paper and put these details in Appendix A.3.
>
> ****
>
> >**Q3**: The effectiveness of the proposed evaluation metric is a function of the evaluation model. Please provide further details on the fine-tuning method and dataset utilized for the evaluation model.
>
> **A3**: We randomly select 800 processed examples from each field for human annotation. As is shown above, this dataset includes the original input (images and questions), ground truth, and responses from evaluation models. To facilitate the annotation process, we design a graphical annotation tool, as shown in the figure below. In this tool, the samples are displayed on the left, while a floating panel on the right allows annotators to score the samples. Annotators can check different criteria, and the final cumulative score is calculated accordingly. Checking a criterion indicates that the sample meets that specific criterion. We have revised the paper and put these details in Appendix A.3.
>
> ****
>
> >**Q4**: To better understand the role that the evaluation model plays in the pipeline, please provide qualitative examples of eval model responses corresponding to Fig 7,9,10,11,12 in the appendix.
>
> **A4**: The evaluation model provides detailed and comprehensive feedback for each example based on the criteria and ground truth. We present the corresponding feedback responses for all the cases. We have revised the paper and put these details in Appendix C (see Figure 8,9,10,11,12,13,14,15).
>
> ****
>
> >**Q5**: Please provide details on how the human annotations were obtained for Table 5 to better support the claim in L110.
>
> **A5**: We design a graphical annotation tool to facilitate the process. Human annotators score the samples by checking options on the floating panel on the right and assigning a final score. Each option represents a specific criterion, and checking the box indicates that the model's response meets that criterion. Specifically, the "emotional support" criterion is checked by default; if unchecked, the score is reduced by 1 point, as explained in Appendix A.3 of the paper. The final score is determined by the number of checked criteria. We strictly follow this scoring process to build our dataset. Using 800 samples from this dataset, we fine-tune the Intern-VL2 model to enhance its scoring capabilities in our task. We then evaluate its performance by comparing the model's scores with human evaluation scores on 200 samples, demonstrating the effectiveness of our scoring model. We have revised the paper and put these details in Appendix A.3.

---

### Meta-Review · Area_Chair_14JV · 2024-12-19

**Metareview:**

This paper introduces MMIE (Massive Multimodal Interleaved Evaluation), a benchmark for evaluating interleaved multimodal comprehension and generation abilities of Multimodal LLMs. MMIE comprises 20K multimodal queries across 12 fields, supporting interleaved text and multi-image inputs and outputs in both multiple-choice and open-ended formats. The authors further propose an automated evaluation metric using a finetuned LLM.  The evaluation dataset is publicly released.

All reviews are positive about the paper, highlighting the originality of a robust evaluation framework for benchmarking the interview input of image and text and its quality. The automatic evaluation metric is also a plus.

**Additional Comments On Reviewer Discussion:**

The reviewers raised the question for more details and potential bias. The authors are engaged in the discussion period and clarified most of the questions.

---

### Decision · Program_Chairs · 2025-01-22

Accept (Oral)